# Don't Throw Away Your Beams: Improving Consistency-based Uncertainties in LLMs via Beam Search

**Ekaterina Fadeeva**[1]    **Maiya Goloburda**[2]    **Aleksandr Rubashevskii**[2]    **Roman Vashurin**[2]

**Artem Shelmanov**[2]    **Preslav Nakov**[2]    **Mrinmaya Sachan**[1]    **Maxim Panov**[2]

[1] ETH Zurich                    [2] MBZUAI

## Abstract

Consistency-based methods have emerged as an effective approach to uncertainty quantification (UQ) in large language models. These methods typically rely on several generations obtained via multinomial sampling, measuring their agreement level. However, in short-form QA, multinomial sampling is prone to producing duplicates due to peaked distributions, and its stochasticity introduces considerable variance in uncertainty estimates across runs. We introduce a new family of methods that employ beam search to generate candidates for consistency-based UQ, yielding improved performance and reduced variance compared to multinomial sampling. We also provide a theoretical lower bound on the beam set probability mass under which beam search achieves a smaller error than multinomial sampling. We empirically evaluate our approach on six QA datasets and find that its consistent improvements over multinomial sampling lead to state-of-the-art UQ performance.

## 1 Introduction

Today, large language models (LLMs) are increasingly being adapted in various safety-critical domains, including medicine (Busch et al., 2025), education (Xing et al., 2025), and law (Shu et al., 2024). This rapid adoption has led to a growing body of work focused on the assessment of the quality and reliability of LLM outputs. An important research direction in this field is Uncertainty Quantification (UQ; Xiao & Wang, 2019; Baan et al., 2023; Xia et al., 2025), which measures the LLM's confidence in their responses.

UQ methods can be categorized into several distinct groups. These include information-based methods that rely on token likelihoods produced by the LLM (Fomicheva et al., 2020); verbalization approaches that prompt models to provide a confidence score (Tian et al., 2023);

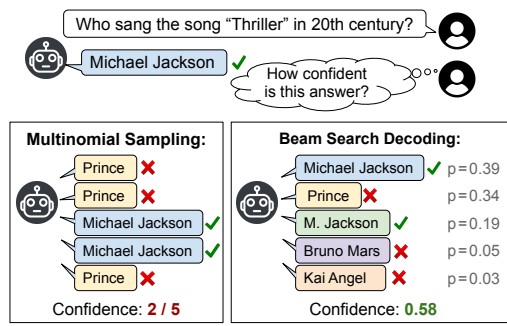

Figure 1: Beam Search vs Multinomial Sampling. Sampling produces multiple identical generations resulting in noisy confidence estimate, while beam search covers top answers from LLM distribution resulting in a better confidence score.

density-based methods that utilize embeddings (Yoo et al., 2022); and last but not least, consistency-based measures that evaluate agreement between sampled outputs (Lin et al., 2024).

Consistency-based UQ methods are of particular interest due to not only their strong performance but also their applicability to black-box settings (Vashurin et al., 2025a).

Correspondence to: `efadeeva@ethz.ch`, `maxim.panov@mbzuai.ac.ae`
Code available at: `https://github.com/IINemo/lm-polygraph/tree/beam-uncertainty`

Moreover, in white-box settings too, it was shown that combining information-based and consistency-based methods yields state-of-the-art performance for a variety of tasks (Kuhn et al., 2023; Duan et al., 2024). A key component of these methods is sampling, which serves as a practical means of approximating the full probability space of all potential model outputs.

Most existing UQ approaches rely on multinomial sampling from the model's output distribution. However, in short-form QA, multinomial sampling is prone to producing similar or even identical generations, due to its bias towards higher-probability tokens during decoding; see Figure 1. Furthermore, since each run produces a different set of candidate outputs, sample-based uncertainty estimates exhibit high variance, undermining their robustness. This limits their effectiveness as a representation of the full output space, especially since, for computational efficiency, studies typically rely on a small number of samples.

To address this problem, we propose computing output consistency based on samples generated using beam search. Beam search facilitates the exploration of alternative decoding paths, which in turn allows one to generate distinct candidate outputs that better capture the model's output space in short-form QA. Our approach includes weighting beam search outputs by their probabilities rather than uniformly, thereby preventing the overrepresentation of low-probability outputs. Particularly, when beam search is employed for decoding, uncertainty estimates are obtained at essentially no additional cost. We show that replacing multinomial sampled outputs with those generated via beam search improves the robustness and accuracy of existing consistency-based methods, as well as hybrid methods relying on both output consistency and token likelihoods.

Our main **contributions** are as follows.

- We identify key limitations of existing consistency-based uncertainty quantification methods based on multinomial sampling; see Section 2.

- We propose a new family of UQ methods that employ an importance-weighted estimator of consistency-based uncertainty with beam search output candidates; see Section 3.

- We provide a distribution-free sufficient condition ensuring that the beam-weighted estimator achieves a lower error than the expected error of the multinomial sampler; see Section 3.2.

- We show that applying a beam search-based estimator to existing consistency-based UQ approaches improves their performance on short-form QA tasks, achieving state-of-the-art results; see Section 4.

## 2 BACKGROUND AND MOTIVATION

### 2.1 LANGUAGE MODEL DECODING

Autoregressive LLMs produce text sequentially, generating one token at a time. At each step $i$, the model samples a token $y_i \sim p(\cdot \mid \mathbf{y}_{<i}, \mathbf{x})$, where $\mathbf{y}_{<i}$ denotes the sequence of previously generated tokens. The probability of generating an output sequence $\mathbf{y}$ is:

$$p(\mathbf{y} \mid \mathbf{x}) = \prod_{i=1}^{|\mathbf{y}|} p(y_i \mid \mathbf{y}_{<i}, \mathbf{x}). \tag{1}$$

At each step, the model outputs a probability distribution over the entire vocabulary $\mathcal{V}$ conditioned on the prompt $\mathbf{x}$ and the partial sequence $\mathbf{y}_{<i}$.

**Decoding strategies.** Since the model defines a probability distribution, a concrete output must be obtained at inference time by applying a decoding strategy. Common decoding strategies include: (i) greedy decoding that selects maximum probability tokens at each step; (ii) multinomial sampling where tokens are drawn according to $p(y_i \mid \mathbf{y}_{<i}, \mathbf{x})$; and (iii) beam search, which maintains the top-$k$ most likely partial sequences at each step. Several other variants of decoding approaches have been proposed, such as top-$p$ nucleus sampling or temperature scaling (Holtzman et al., 2020; Vijayakumar et al., 2018). Each decoding strategy offers different trade-offs between output quality and diversity.

## 2.2 Uncertainty Quantification for LLMs

The objective of uncertainty quantification is to measure the level of uncertainty introduced by LLM when generating output sequence $\mathbf{y}_*$ conditioned on input sequence $\mathbf{x}$, denoted by $U(\mathbf{y}_* \mid \mathbf{x})$. Existing approaches to UQ can be broadly categorized into three main groups.

*Information-based* methods rely on a single forward pass of the model and compute statistics over the token-level probability distributions to quantify uncertainty. Examples include Sequence Probability, Mean Token Entropy, Perplexity (Fomicheva et al., 2020), and CCP (Fadeeva et al., 2024).

*Reflexive* methods query the model directly about its confidence in a generated answer using specially designed prompts. A representative example is *P(True)* (Kadavath et al., 2022), which measures the probability that the model outputs "True" when asked whether its generated answer $\mathbf{y}_*$ is correct.

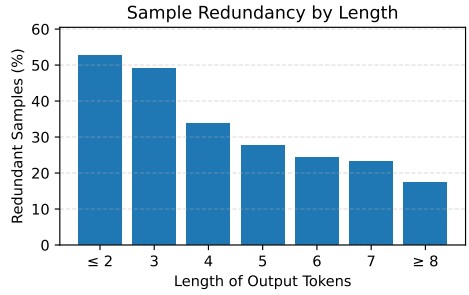

Figure 2: Mean percentage of redundant samples (i.e., outputs already seen among earlier generations) as a function of greedy output length. Results were obtained from 2,000 questions from the TriviaQA dataset using the Gemma 3 4B base model and 10 candidate generations. Redundancy is especially high for short answers, leading to wasted computation.

*Sampling-based* methods draw multiple samples from the model's output distribution and evaluate their semantic or lexical similarity to assess uncertainty. Lexical Similarity (Fomicheva et al., 2020) computes mean pairwise similarity between generated texts; other examples include Semantic Entropy (Kuhn et al., 2023), SAR (Duan et al., 2024), and black-box uncertainty measures from (Lin et al., 2024).

**Consistency-based UQ methods.** A notable subset of sampling-based methods is *consistency-based UQ* (Vashurin et al., 2025b). These methods estimate uncertainty *with respect to a particular generated output* $\mathbf{y}_* \sim p(\cdot \mid \mathbf{x})$, rather than the overall uncertainty of the model's predictive distribution for the input $\mathbf{x}$. This distinction makes consistency-based UQ particularly suited for evaluating confidence in a specific prediction rather than overall model uncertainty, and Vashurin et al. (2025b) empirically demonstrate that such methods outperform other sampling-based approaches in practice.

Let us consider the most straightforward consistency–based method for predictive uncertainty quantification: measuring how semantically different alternative generations are from the produced answer $\mathbf{y}_*$. We refer to this score as *Dissimilarity* and formalize it as the expected semantic dissimilarity between the produced answer $\mathbf{y}_*$ and *all* potential alternatives drawn from the model:

$$U_D(\mathbf{y}_* \mid \mathbf{x}) = \mathbb{E}_{\mathbf{y} \sim p(\cdot \mid \mathbf{x})} \big[ 1 - s(\mathbf{y}, \mathbf{y}_*) \big]. \tag{2}$$

Here, $s(\mathbf{y}', \mathbf{y}'') \in [0, 1]$ is a function that measures semantic similarity between two generations $\mathbf{y}'$ and $\mathbf{y}''$. A higher value of $U_D(\mathbf{y}_* \mid \mathbf{x})$ indicates lower consistency between the chosen answer and alternative candidate outputs, and thus reflects greater predictive uncertainty.

The corresponding Monte Carlo estimator introduced by (Lin et al., 2024) draws $M$ i.i.d. samples $\mathbf{y}^{(1)}, \ldots, \mathbf{y}^{(M)} \sim p(\cdot \mid \mathbf{x})$ and computes uncertainty in the following way:

$$\widehat{U}_D^{MC}(\mathbf{y}_* \mid \mathbf{x}) = \frac{1}{M} \sum_{i=1}^{M} \big( 1 - s(\mathbf{y}^{(i)}, \mathbf{y}_*) \big). \tag{3}$$

**Challenges of consistency-based UQ methods.** A natural intuition is that, for consistency-based methods, samples should be generated in a distinct, high-probability, and stable manner. Most existing methods use multinomial sampling, which, especially for shorter generations and small sample sizes, does not satisfy these criteria.

Figure 2 shows the effect of multinomial sampling on the percentage of duplicates depending on the length of generations. The resulting samples contain many duplicates, with the issue being particularly pronounced for shorter generations, where 30–50% of the outputs are duplicates.

This not only contributes to wasted computation, but also leads to high variance estimates. Moreover, drawing $M$ full generations solely for uncertainty estimation can be costly.

Thus, while multinomial sampling is widely used, it does not best serve the goals of consistency-based uncertainty estimation.

# 3 UNCERTAINTY QUANTIFICATION BASED ON CONSISTENCY OF BEAM SEARCH CANDIDATES

To address the problems outlined above, we propose to utilize an alternative decoding strategy for generating candidate outputs: beam search. Beam search (i) guarantees distinct candidate outputs, (ii) reduces variance (see Section 3.2) and (iii) provides uncertainty estimates essentially "for free" as the beam already provides a distribution over candidate outputs.

## 3.1 REPLACING MULTINOMIAL SAMPLING

A simple way to approximate dissimilarity from beam-generated candidates would be to reuse equation (3), treating the beam outputs as if they were drawn uniformly. While this offers a plausible alternative, treating the candidates produced by beam search in a uniform manner would overemphasize lower-probability outputs. To better reflect the model distribution while avoiding repeated multinomial draws, we form a probability-weighted estimator over the beam set.

For this purpose, we use beam search with width $M$ to obtain distinct candidates $\mathcal{B}_M(\mathbf{x}) = \{\mathbf{b}^{(1)}, \ldots, \mathbf{b}^{(M)}\}$ and their sequence probabilities $\{p(\mathbf{b}^{(i)} \mid \mathbf{x})\}_{i=1}^{M}$. To perform an estimation of $U_D(\mathbf{y}_* \mid \mathbf{x})$ in equation (2) with the help of samples $b^{(i)}$, one needs to perform importance weighting. Thus, we define the restricted (top-$M$) normalized masses $w_i$ as:

$$w_i = \frac{p(\mathbf{b}^{(i)} \mid \mathbf{x})}{\sum_{j=1}^{M} p(\mathbf{b}^{(j)} \mid \mathbf{x})}, \qquad i = 1, \ldots, M. \tag{4}$$

The resulting importance-weighted estimator of equation (2) is

$$\widehat{U}_D^b(\mathbf{y}_* \mid \mathbf{x}) = \sum_{i=1}^{M} w_i \big(1 - s(\mathbf{b}^{(i)}, \mathbf{y}_*)\big). \tag{5}$$

This top-$M$ truncation introduces a small bias relative to full multinomial sampling but typically reduces variance and duplication on peaked distributions, yielding more stable estimates per unit budget. In the next section we are going to explore the benefits of beam search-based estimator $\widehat{U}_D^b(\mathbf{y}_* \mid \mathbf{x})$ from a theoretical perspective.

## 3.2 THEORETICAL ANALYSIS

We compare the multinomial Monte Carlo estimator $\widehat{U}_D^{MC}$ (3) with the beam-weighted estimator $\widehat{U}_D^b$ (5) for the dissimilarity $U_D$ defined in equation (2).

**Theorem 1** (Comparison condition for beam-weighted and Monte Carlo estimators).
*Let $\mathcal{B}_M(\mathbf{x}) = \{\mathbf{b}^{(1)}, \ldots, \mathbf{b}^{(M)}\}$ be the beam set, $m_{\mathcal{B}} = \sum_{i=1}^{M} p(\mathbf{b}^{(i)} \mid \mathbf{x})$ be its total probability mass, and define $\mu_{\mathcal{B}}$ and $\mu_{\overline{\mathcal{B}}}$ as dissimilarity inside and outside the beam set $\mathcal{B}_M$ correspondingly:*

$$\mu_{\mathcal{B}} = \mathbb{E}_{\mathbf{y} \sim p(\cdot \mid \mathbf{x})}\left[1 - s(\mathbf{y}, \mathbf{y}_*) \mid \mathbf{y} \in \mathcal{B}_M(\mathbf{x})\right], \qquad \mu_{\overline{\mathcal{B}}} = \mathbb{E}_{\mathbf{y} \sim p(\cdot \mid \mathbf{x})}\left[1 - s(\mathbf{y}, \mathbf{y}_*) \mid \mathbf{y} \notin \mathcal{B}_M(\mathbf{x})\right].$$

*Then the beam-weighted estimator $\widehat{U}_D^b$ achieves smaller mean-squared error than the Monte Carlo estimator $\widehat{U}_D^{MC}$ whenever*

$$(1 - m_{\mathcal{B}})\big|\mu_{\mathcal{B}} - \mu_{\overline{\mathcal{B}}}\big| < \sigma/\sqrt{M}, \tag{6}$$

*where $\sigma^2 = \mathrm{Var}_{\mathbf{y} \sim p(\cdot \mid \mathbf{x})}(1 - s(\mathbf{y}, \mathbf{y}_*))$. The corresponding distribution-free sufficient condition is*

$$m_{\mathcal{B}} > 1 - \frac{1}{2\sqrt{M}}. \tag{7}$$

*Proof.* The Monte Carlo estimator averages $M$ i.i.d. samples $\mathbf{y}^{(i)} \sim p(\cdot \mid \mathbf{x})$, so it is unbiased with $\mathbb{E}[\widehat{U}_D^{MC}] = U_D(\mathbf{y}_* \mid \mathbf{x})$ and $\mathrm{MSE}(\widehat{U}_D^{MC}) = \mathrm{Var}(\widehat{U}_D^{MC}) = \sigma^2/M$. By Popoviciu's inequality, any random variable supported on $[0,1]$ has variance at most $1/4$, hence $\sigma^2 \leq 1/4$.

By the law of total expectation, the true dissimilarity $U_D$ decomposes as:

$$U_D(\mathbf{y}_* \mid \mathbf{x}) = m_{\mathcal{B}}\mu_{\mathcal{B}} + (1 - m_{\mathcal{B}})\mu_{\overline{\mathcal{B}}}, \qquad \widehat{U}_D^b = \mu_{\mathcal{B}},$$

so squared error of the beam-weighted estimator $\widehat{U}_D^b$ is deterministic:

$$\mathrm{SE}(\widehat{U}_D^b) = \left(\widehat{U}_D^b - U_D\right)^2 = (1 - m_{\mathcal{B}})^2\left(\mu_{\mathcal{B}} - \mu_{\overline{\mathcal{B}}}\right)^2.$$

Beam-weighted estimation is therefore more accurate whenever

$$(1 - m_{\mathcal{B}})^2\left(\mu_{\mathcal{B}} - \mu_{\overline{\mathcal{B}}}\right)^2 < \sigma^2/M,$$

which yields the stated condition (6). A distribution-free bound (7) follows from $\left|\mu_{\mathcal{B}} - \mu_{\overline{\mathcal{B}}}\right| \leq 1$ and $\sigma^2 \leq 1/4$. □

From Theorem 1, beam-weighted estimator is more accurate than Monte Carlo estimator whenever total beam probability mass $m_{\mathcal{B}}$ exceeds $1 - \frac{1}{2\sqrt{M}}$. For $M = 10$, the threshold is $m_{\mathcal{B}} > 0.842$. Thus, when the top-10 beam hypotheses capture at least $\sim 84\%$ of the model's probability mass, beam search provides a lower-error estimator than multinomial sampling with the same sample budget.

In practice, *this condition is frequently satisfied.* On the TriviaQA dataset, Figure 3 shows that $22.7\%$ of examples meet the sufficient condition overall, and up to 30-40% for very short generations ($\leq 3$ output tokens), where proba-

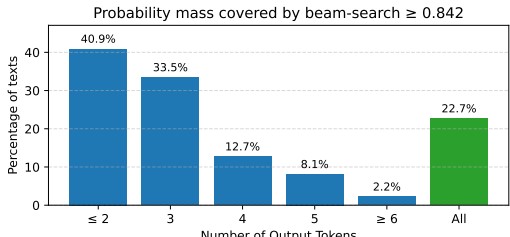

Figure 3: Percentage of texts meeting the sufficient condition (Theorem 1). Results are based on 2,000 TriviaQA questions, Gemma 3 4B base and $M = 10$. The green "All" bar shows the overall percentage across all lengths.

bility mass is highly concentrated on the top beams. When the inside-outside gap $\delta = |\mu_{\mathcal{B}} - \mu_{\overline{\mathcal{B}}}| < 1$, the break-even requirement (6) relaxes to $(1 - m_{\mathcal{B}})\delta < \sigma/\sqrt{M}$, allowing beam search to outperform even when $m_{\mathcal{B}} < 0.842$. Although $\mu_{\overline{\mathcal{B}}}$ is not directly computable due to the combinatorial output space, our experiments consistently show beam search outperforming multinomial sampling, suggesting that $\delta$ is modest in practice and that the effective threshold is often lower than $0.842$.

### 3.3 ADAPTING OTHER UQ METHODS TO BEAM SEARCH

In a similar manner, other consistency-based methods can be adapted to utilize beam search-based samples in their formulation.

**Eccentricity.** Eccentricity is a method introduced by Lin et al. (2024). Unlike dissimilarity, which uses only the similarities between the produced answer $\mathbf{y}_*$ and each alternative sample, Eccentricity aggregates the *joint* pairwise relationships among all samples.

In this method, we first construct a similarity matrix of size $(M + 1) \times (M + 1)$ for the $M$ samples and the produced answer $\mathbf{y}^{(M+1)} = \mathbf{y}_*$:

$$W_{ij} = s\left(\mathbf{y}^{(i)}, \mathbf{y}^{(j)}\right), \quad 1 \leq i, j \leq M + 1. \tag{8}$$

Then we compute the degree matrix $D$:

$$D_{ij} = \begin{cases} \sum\limits_{k=1}^{M+1} W_{ik}, & i = j, \\ 0, & i \neq j, \end{cases} \tag{9}$$

and obtain the eigendecomposition of the Graph Laplacian $L = I - D^{-1/2}WD^{-1/2}$, yielding eigenpairs $\{\lambda_i, \mathbf{u}_i\}_{i=1}^{M+1}$.

Smaller eigenvalues (close to zero) capture meaningful semantic structure, whereas larger eigenvalues tend to reflect noise. We therefore retain the eigenvectors whose eigenvalues satisfy $\lambda_i < \alpha$, yielding $K$ vectors in total; $K$ is thus determined by the threshold $\alpha > 0$.

Semantic embeddings are formed as $\mathbf{v}_j = [\mathbf{u}_{1j}, \mathbf{u}_{2j}, \ldots, \mathbf{u}_{Kj}]$. For $1 \leq j \leq M$, $\mathbf{v}_j$ represents the embedding of $\mathbf{y}^{(j)}$, and $\mathbf{v}_* = \mathbf{v}_{M+1}$ corresponds to $\mathbf{y}_*$. The confidence score is the distance between the embedding of the produced answer and the mean embedding of the samples:

$$\widehat{U}_{Ecc}(\mathbf{y}_* \mid \mathbf{x}) = \left\| \mathbf{v}_* - \frac{1}{M} \sum_{i=1}^{M} \mathbf{v}_i \right\|_2^2, \tag{10}$$

where higher values indicate higher uncertainty.

With beam-generated candidates, we weight embeddings by the normalized masses $w_i$ from equation (4) to better reflect the model distribution while avoiding duplicate generations:

$$\widehat{U}_{Ecc}^b(\mathbf{y}_* \mid \mathbf{x}) = \left\| \mathbf{v}_*^b - \sum_{i=1}^{M} w_i \mathbf{v}_i^b \right\|_2^2. \tag{11}$$

**CoCoA.** A white-box approach CoCoA (Vashurin et al., 2025b) combines a model probabilities-based uncertainty with the sample-consistency signal:

$$\widehat{U}_{CoCoA}(\mathbf{y}_* \mid \mathbf{x}) = u(\mathbf{y}_* \mid \mathbf{x}) \cdot \widehat{U}_D^{MC}(\mathbf{y}_* \mid \mathbf{x}) = u(\mathbf{y}_* \mid \mathbf{x}) \cdot \frac{1}{M} \sum_{i=1}^{M} \big(1 - s(\mathbf{y}^{(i)}, \mathbf{y}_*)\big), \tag{12}$$

where $u(\mathbf{y} \mid \mathbf{x})$ is a model-based uncertainty measure for the sequence (e.g., $-\log p(\mathbf{y} \mid \mathbf{x})$).

For a beam-weighted estimator, we utilize (5) as sample-consistency signal:

$$\widehat{U}_{CoCoA}^b(\mathbf{y}_* \mid \mathbf{x}) = u(\mathbf{y}_* \mid \mathbf{x}) \cdot \widehat{U}_D^b(\mathbf{y}_* \mid \mathbf{x}). \tag{13}$$

**Eigenvectors Dissimilarity.** Both Dissimilarity and Eccentricity produce confidence scores for the generated answer $\mathbf{y}_*$. Dissimilarity compares $\mathbf{y}_*$ to each sample using the base similarity function $s$, while Eccentricity measures the distance from $\mathbf{y}_*$ to the centroid in the Laplacian embedding space; see equation (10). To bridge these views, we measure dissimilarity within the embedding space itself, averaging the distances from the embedding of $\mathbf{y}_*$ to the embeddings of individual samples. This retains the joint-pairwise smoothing of Eccentricity and also reflects the variance among samples, rather than only the centroid. The sampling-based estimate is

$$\widehat{U}_{EigVecD}(\mathbf{y}_* \mid \mathbf{x}) = \frac{1}{M} \sum_{i=1}^{M} \left\| \mathbf{v}_* - \mathbf{v}_i \right\|_2^2, \tag{14}$$

and the beam-guided, probability-weighted version is

$$\widehat{U}_{EigVecD}^b(\mathbf{y}_* \mid \mathbf{x}) = \sum_{i=1}^{M} w_i \left\| \mathbf{v}_*^b - \mathbf{v}_i^b \right\|_2^2, \tag{15}$$

where the embeddings $\mathbf{v}_i$ (and $\mathbf{v}_i^b$) are obtained from the Graph Laplacian as in Eccentricity, and $w_i$ are the normalized masses from equation (4). This estimator increases both when $\mathbf{y}_*$ moves away from the bulk and when the samples themselves are more dispersed; by contrast, Eccentricity focuses on the single distance to the weighted centroid.

## 4 EXPERIMENTS

### 4.1 EXPERIMENTAL SETUP

**Datasets.** We evaluate our approach on six QA datasets in total. Those include two closed-book datasets: *TriviaQA* (Joshi et al., 2017) and *Web Questions* (Berant et al., 2013), two open-book datasets: *CoQA* (Reddy et al., 2019) and *HotpotQA* (Yang et al., 2018) and two multiple-choice datasets: *CommonsenceQA* (Talmor et al., 2019) and *ARC-Challenge* (Clark et al., 2018). For each dataset, we randomly sampled several questions from the test set. The statistics for those datasets are available in Table 1. Prompt details and examples of questions are provided in Appendix C.

Table 1: Test dataset settings and statistics.

| | Closed-Book QA | | Open-Book QA | | Multiple Choice | |
| --- | --- | --- | --- | --- | --- | --- |
| | TriviaQA | Web Questions | CoQA | HotpotQA | Common senceQA | ARC-Challenge |
| # Questions | 2000 | 1490 | 2000 | 2000 | 1221 | 447 |
| # few-shot examples | 5 | 5 | all preceding | 0 | 2 | 2 |
| Max new tokens | 20 | 20 | 20 | 20 | 10 | 20 |

Table 2: Summary of baseline UQ methods.

| Category | Uncertainty Quantification Method |
| --- | --- |
| Information-based | Sequence Probability (Prob) 
 Mean Token Entropy (MTE) 
 Perplexity 
 CCP (Fadeeva et al., 2024) |
| Reflexive | P(True) (Kadavath et al., 2022) |
| Sampling-based | Semantic Entropy (Kuhn et al., 2023) 
 Shifting Attention to Relevance (SAR) (Duan et al., 2024) 
 Lexical Similarity (Fomicheva et al., 2020) 
 Sum of Eigenvalues of Laplacian (EigValLaplacian) (Lin et al., 2024) 
 Number of Semantic Sets (NumSemSets) (Lin et al., 2024) |

**Models.** We use base and instruct versions of 3 models: Gemma 3 4B (Team, 2025a), Llama 3.1 8B (Dubey et al., 2024), and Qwen 3 8B (Team, 2025b).

**Metrics.** Following best uncertainty benchmarking practices (Vashurin et al., 2025a), we adopt the Prediction–Rejection Ratio (PRR) as our primary evaluation metric. Consider a test dataset $\mathcal{D} = \{(\mathbf{x}_j, \mathbf{t}_j)\}$, where $\mathbf{t}_j$ denotes target output. Then, we can obtain an output $\mathbf{y}_j^*$ generated by an LLM for input $\mathbf{x}_j$ and the associated uncertainty score $u_j = U(\mathbf{y}_j^* \mid \mathbf{x}_j)$.

Based on these we can build rejection curve that captures how the average quality $Q(\mathbf{y}_j^*, \mathbf{t}_j)$ over all $\{(\mathbf{y}_j^*, \mathbf{t}_j) \colon u_j < \tau\}$ changes with the rejection threshold $\tau$. An oracle rejection curve can be defined by substituting $u_j = -Q(\mathbf{y}_j^*, \mathbf{t}_j)$, giving best possible rejection order where lowest-quality outputs are rejected first. A baseline for rejection can be obtained by rejecting outputs uniformly at random. PRR is then defined as the ratio of the area between UQ rejection curve and a random rejection baseline to the area between oracle rejection and the same random baseline:

$$PRR = \frac{\text{AUC}_{\text{unc}} - \text{AUC}_{\text{rnd}}}{\text{AUC}_{\text{oracle}} - \text{AUC}_{\text{rnd}}}. \tag{16}$$

A higher PRR indicates a more effective uncertainty score. Following Vashurin et al. (2025a), we use AlignScore (Zha et al., 2023) as the quality metric $Q$. While PRR serves as our main evaluation measure, we additionally report ROC-AUC and PR-AUC in Appendix D.2.

**Baselines.** We evaluate four main methods, Dissimilarity, Eccentricity, Eigenvectors Dissimilarity, and CoCoA, under multinomial sampling and their beam-guided, probability-weighted variants. For CoCoA, we consider both *CocoaMSP* based on unnormalized log-probability:

$$u(\mathbf{y}_* \mid \mathbf{x}) = -\log p(\mathbf{y}_* \mid \mathbf{x}), \tag{17}$$

and *CocoaPPL* based on perplexity:

$$u(\mathbf{y}_* \mid \mathbf{x}) = -\frac{1}{|\mathbf{y}_*|} \log p(\mathbf{y}_* \mid \mathbf{x}). \tag{18}$$

In addition, we compare against several state-of-the-art UQ baselines summarized in Table 2, using implementations from LM-Polygraph (Fadeeva et al., 2023). The simplest baseline, *Sequence Probability*, calculates $-\log p(\mathbf{y}_* \mid \mathbf{x})$. For detailed descriptions of other methods see Appendix E.

Table 3: PRR (↑ is better) averaged over 6 datasets. For each model, the top-1 method is **bold** and the second-best is underlined. For beam-guided variants, we mark ↑ when the variant improves over its original multinomial-sampling counterpart.

| Method | Llama 3.1 8B base | Llama 3.1 8B instruct | Gemma 3 4B base | Gemma 3 4B instruct | Qwen 3 8B base | Qwen 3 8B instruct |
|---|---|---|---|---|---|---|
| *Baseline UQ Methods* | | | | | | |
| Prob | .410 ± .019 | .344 ± .031 | .471 ± .023 | .292 ± .022 | .376 ± .03 | .289 ± .067 |
| MTE | .422 ± .016 | .364 ± .026 | .476 ± .022 | .317 ± .028 | .407 ± .032 | .297 ± .064 |
| Perplexity | .452 ± .02 | .323 ± .027 | .525 ± .024 | .288 ± .025 | .372 ± .03 | .276 ± .058 |
| CCP | .401 ± .02 | .364 ± .029 | .492 ± .022 | .331 ± .026 | .355 ± .034 | .291 ± .06 |
| SAR | .352 ± .02 | .385 ± .029 | .386 ± .026 | .239 ± .024 | .363 ± .033 | .292 ± .052 |
| P(True) | .015 ± .023 | .072 ± .03 | .093 ± .026 | -.096 ± .024 | .110 ± .03 | -.114 ± .055 |
| Semantic Entropy | .414 ± .019 | .376 ± .025 | .401 ± .023 | .293 ± .024 | .319 ± .031 | .299 ± .058 |
| Lexical Similarity | .411 ± .02 | .366 ± .029 | .426 ± .025 | .247 ± .023 | .425 ± .034 | .237 ± .055 |
| EigValLaplacian | .426 ± .016 | .371 ± .028 | .437 ± .03 | .233 ± .025 | .406 ± .03 | .265 ± .056 |
| NumSemSets | .396 ± .018 | .319 ± .031 | .418 ± .024 | .238 ± .023 | .365 ± .033 | .253 ± .052 |
| *Consistency-based UQ: multinomial vs. beamsearch versions* | | | | | | |
| Dissimilarity | .505 ± .018 | .379 ± .028 | .630 ± .021 | .206 ± .019 | .477 ± .037 | .327 ± .066 |
| Dissimilarity + beamsearch | **.543** ↑± .019 | .417 ↑± .026 | **.650** ↑± .022 | .252 ↑± .022 | .478 ↑± .031 | .355 ↑± .062 |
| Eccentricity | .453 ± .016 | .368 ± .029 | .563 ± .021 | .231 ± .025 | .396 ± .035 | .251 ± .058 |
| Eccentricity + beamsearch | .505 ↑± .017 | .397 ↑± .029 | .603 ↑± .023 | .285 ↑± .024 | .410 ↑± .03 | .345 ↑± .061 |
| EigVecDissimilarity | .463 ± .019 | .370 ± .028 | .561 ± .026 | .236 ± .025 | .425 ± .035 | .256 ± .051 |
| EigVecDissimilarity + beamsearch | .510 ↑± .021 | .414 ↑± .028 | .598 ↑± .022 | .301 ↑± .019 | .450 ↑± .033 | **.376** ↑± .057 |
| CocoaMSP | .505 ± .018 | .404 ± .025 | .587 ± .023 | .314 ± .024 | .461 ± .031 | .334 ± .054 |
| CocoaMSP + beamsearch | .521 ↑± .019 | **.426** ↑± .024 | .615 ↑± .021 | **.345** ↑± .026 | .473 ↑± .03 | .347 ↑± .061 |
| CocoaPPL | .523 ± .017 | .397 ± .026 | .628 ± .024 | .312 ± .023 | .461 ± .034 | .327 ± .055 |
| CocoaPPL + beamsearch | .536 ↑± .02 | .412 ↑± .027 | .649 ↑± .026 | .339 ↑± .021 | .461 ↑± .035 | .337 ↑± .057 |

All experiments use $M = 10$ candidates for both multinomial sampling and beam search. We adopt the entailment probability from the DeBERTa-large model fine-tuned on the MNLI task (He et al., 2021) for similarity function $s$, following Lin et al. (2024).

## 4.2 RESULTS AND DISCUSSION

Table 3 presents PRR results for six models, averaged over six datasets. Across all models, incorporating beam search consistently improves the performance of consistency-based uncertainty scores. Moreover, in almost every case, beam search–based methods achieve either the best or second-best PRR compared to both baselines and the original consistency-based approaches. In particular, Dissimilarity + Beam Search achieves the best PRR scores for all base models and the second-best scores for Llama 3.1 8B instruct and Qwen 3 8B instruct. Similarly, CocoaMSP + Beam Search achieves the best results for Llama 3.1 8B instruct and Gemma 3 4B instruct, while CocoaPPL + Beam Search ranks second-best for Llama 3.1 8B base, Gemma 3 4B base, and Gemma 3 4B instruct. We further provide separate results for each dataset in Appendix D.3.

## 4.3 ABLATIONS

In this section, we study sensitivity to (i) the number of candidates $M$, (ii) output length, and (iii) rejection rate in PRR curves.

### 4.3.1 EFFECT OF SAMPLE COUNT

We vary the sample count $M \in \{1, \ldots, 15\}$ for Dissimilarity, Eccentricity, and EigVecDissimilarity under multinomial sampling and beam search. Figure 5 shows that beam search generally achieves higher PRR across all budgets $M \geq 2$. Notably, beam search reaches high PRR at small budgets (3-5 samples) and saturates quickly, while multinomial sampling improves more gradually and remains below beam search throughout.

For $M = 1$, beam search reduces to greedy decoding, causing Dissimilarity to be nearly zero because it compares two identical greedy outputs. In contrast, the sampling variant compares greedy decoding to a stochastic sample, yielding a more informative value.

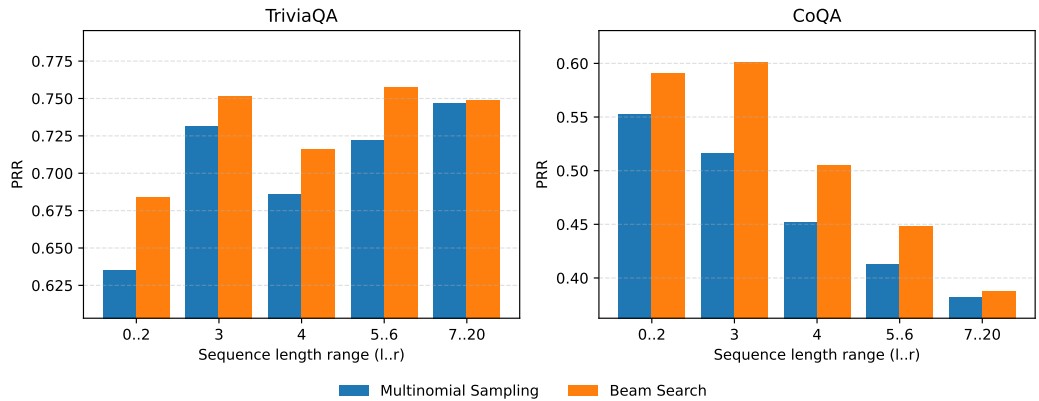

Figure 4: PRR (↑ is better) for Dissimilarity under beam search (with probability weights) vs. multinomial sampling, for different output lengths. Each dataset (TriviaQA, CoQA) with Gemma 3 4B base is partitioned into five approximately equal-size bins token length of greedy output.

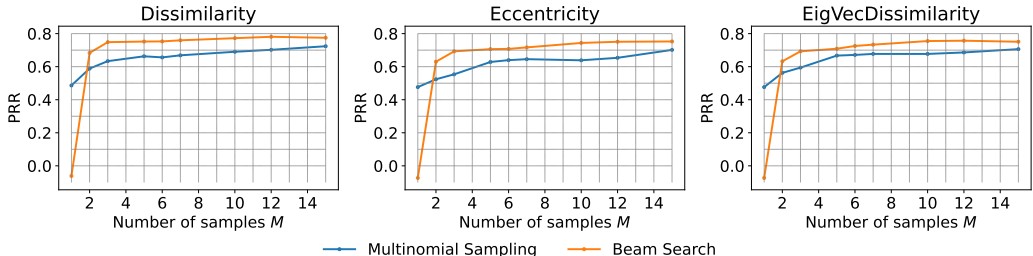

Figure 5: PRR (↑ is better) as a function of the number of candidates $M$ on TriviaQA with Gemma 3 4B base. Each panel reports one estimator (Dissimilarity, Eccentricity, EigVecDissimilarity). Curves compare multinomial sampling and beam search (with probability weights from equation (4)).

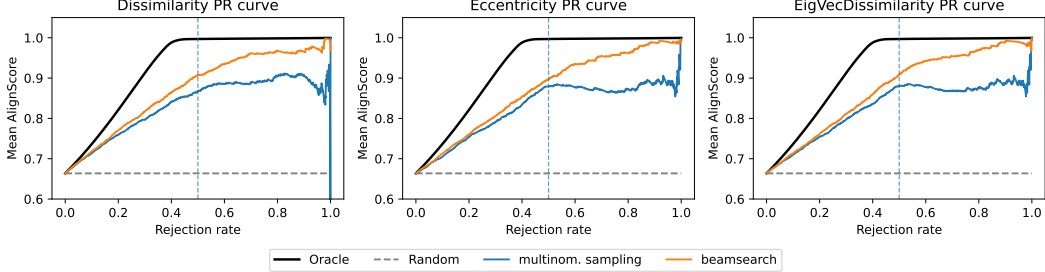

Figure 6: Prediction-Rejection curves for *Dissimilarity*, *Eccentricity*, and *EigVecDissimilarity* on TriviaQA with Llama 3.1 8B base, comparing multinomial sampling (blue) and beam search with weights (orange). Oracle (black) and random (gray dashed) baselines are shown. The vertical dashed line marks the maximum rejection rate used in AUC calculations.

### 4.3.2 EFFECT OF OUTPUT LENGTH

Beam-guided estimators outperform sampling-based ones most clearly when generations are short. As shown earlier in Figure 2, duplicate rates under multinomial sampling are high for 2-4 tokens ($\sim$ 30–50%) and drop to $\sim$ 17% for outputs of 8+ tokens. To quantify the impact, we compute PRR for Dissimilarity using beam search (with weights from equation (4)) and multinomial sampling (no weights) across five length bins of approximately equal size on TriviaQA and CoQA with Gemma 3 4B base; see Figure 4. Within each bin, beam search consistently beats multinomial sampling for short outputs; the gap narrows and becomes negligible for lengths of about 7 tokens and above, where duplication is less pronounced.

### 4.3.3 Prediction-Rejection Curves

Figure 6 compares full Prediction-Rejection curves for Dissimilarity, Eccentricity, and EigVecDissimilarity on TriviaQA with Llama 3.1 8B base. Across all estimators, beam search consistently dominates multinomial sampling for nearly the entire rejection range. The becomes increasingly pronounced as the rejection rate grows, where beam-guided estimates remain stable while multinomial ones flatten or even degrade. This indicates that beam search is especially beneficial in the high-rejection regime, where distinguishing between stronger and weaker candidates is most critical.

### 4.3.4 Additional Ablations

Additional ablations are deferred to the appendices: Appendix A.1 compares candidate–generation strategies including Diverse Beam Search, temperature sampling, and a hybrid multinomial–beam sampling. Appendix A.2 investigates restricted-mass normalization and shows that introducing a small probability floor $\epsilon$ can stabilize the weighting of low-mass beams. Appendix A.3 evaluates other sampling-based objectives (Semantic Entropy, Degree Matrix) under beam generation with probability-weighted formulations. Appendix D.1 examines top-1 beam decode as the produced answer $\mathbf{y}_*$ (instead of greedy), a natural choice when beam search has already been run.

## 5 Related Work

**Consistency-based uncertainty estimation.** In a black-box setting, consistency-based methods are especially relevant, as they do not require access to the model internals. Lin et al. (2024) introduce several methods that estimate confidence based on a similarity matrix, where each entry represents the similarity between a pair of sampled generations. Fomicheva et al. (2020) present Lexical Similarity, a metric that evaluates the average similarity of words or phrases between each pair of responses. In a white-box setting, consistency signals can be combined with model token-probabilities-based confidence. These hybrid methods, such as Semantic Entropy (Kuhn et al., 2023), CoCoA (Vashurin et al., 2025b) and SAR (Duan et al., 2024) explore different ways of combining these signals and achieve state-of-the-art performance. However, these works are primarily concerned with the introduction of new methods for uncertainty quantification and use multinational sampling as a way to approximate a variety of consistency-based measures.

**Uncertainty and decoding.** There were some efforts focused on examining the interaction between decoding strategies and uncertainty quantification. In particular, Hashimoto et al. (2025) explores the impact of decoding strategies on the performance of token probabilities-based UQ methods, namely Sequence Probability and Mean Token Entropy. The authors find that these scores produced with beam search can sometimes under perform compared to greedy or contrastive search. While this work offers interesting insights, no experiments with stochastic decoding strategies or non-likelihood based methods were conducted. Conversely, other research focused on making the decoding itself uncertainty-aware. For example, Daheim et al. (2025) propose Minimum Bayes Risk (MBR) decoding, which incorporates model uncertainty into the MBR objective for improved generation quality. Garces Arias et al. (2024) and Lee et al. (2025) incorporate uncertainty into contrastive search decoding. Lastly, Ding et al. (2025) combines global entropy trends and local deviations to guide a self-adaptive decoding. These works integrate uncertainty into the decoding process to improve the quality of the generation, rather than improving the performance of the uncertainty itself. Although some uncertainty-aware decoding methods have also demonstrated improved uncertainty quantification performance, they are generally not evaluated with consistency-based metrics.

## 6 Conclusion

We present a new family of uncertainty quantification methods for LLMs that employ a beam-weighted estimator of consistency-based uncertainty. Compared to multinomial sampling, commonly used in existing approaches, our method yields lower variance in dissimilarity and greater diversity of candidate answers. We also derive a theoretical lower bound on the beam set probability mass under which the error of the multinomial Monte Carlo estimator is guaranteed to be larger. Finally, we evaluate our approach on six QA datasets and six different models, demonstrating state-of-the-art performance.

LIMITATIONS

Although our method provides an improvement over existing consistency-based estimators, several important considerations remain. First, we evaluated our methods in white-box settings, as they require access to the model's probability distributions. Nonetheless, we argue that developing methods tailored for white-box settings continues to be of great importance given their continued relevance and usage. Moreover, the methods could be extended to the black-box settings using empirical probability estimates.

Second, our experiments are limited to short-form QA datasets, and the generalizability of our findings to longer-form generation remains an open question.

Lastly, our implementation and evaluation relies on existing neural metrics: AlignScore is used to score the quality of the generation, and pre-trained NLI model is utilized as a measure of consistency. Although widely used in previous work, certain more specialized tasks might require different sample similarity measures and quality metrics.

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

# A ABLATION STUDIES

## A.1 DIFFERENT SAMPLING STRATEGIES

This section studies how the proposed estimators behave under different sample generation strategies. In addition to multinomial sampling and beam search settings, we evaluate three additional families.

**Diverse beam search.** We generate $M = 10$ candidates using a diverse beam search (Vijayakumar et al., 2018) with group penalties $\lambda \in \{0.5, 1.0, 1.5, 2.0\}$ and group counts that split the ten candidates into $G \in \{2, 5\}$ groups. As in the main beam setup, we apply the same self-normalized probability weights $w_i$ from equation (4).

**Temperature sampling with importance weights.** For different temperatures $T$, we draw $M = 10$ samples with temperature sampling $\{\mathbf{y}_T^{(i)}\}_{i=1}^M$ and re-weight them via self-normalized importance weights

$$w_i^T = \frac{p\big(\mathbf{y}_T^{(i)} \mid \mathbf{x}\big)^{1-1/T}}{\sum_{j=1}^M p\big(\mathbf{y}_T^{(j)} \mid \mathbf{x}\big)^{1-1/T}}. \tag{19}$$

**Hybrid multinomial–beam search.** We also consider a joint strategy: first draw $B$ beam candidates, then draw the remaining $M - B$ candidates via multinomial sampling while excluding the beam results. Beam candidates use autoregressive probability weights, and the residual probability mass is distributed uniformly over the multinomial samples. Let $\{\mathbf{b}^{(i)}\}_{i=1}^B$ be the beam outputs and $\{\mathbf{y}^{(j)}\}_{j=B+1}^M$ the multinomial samples (with beam sequences masked out). We assign weights

$$w_i^H = p\big(\mathbf{b}^{(i)} \mid \mathbf{x}\big), \quad i = 1, \ldots, B, \qquad w_j^H = \frac{1 - \sum_{i=1}^B p\big(\mathbf{b}^{(i)} \mid \mathbf{x}\big)}{M - B}, \quad j = B + 1, \ldots, M, \tag{20}$$

so that $\sum_{i=1}^M w_i^H = 1$. We test $B \in \{1, \ldots, 9\}$.

Evaluations use a subset of 500 examples from TriviaQA and CoQA with two base models, Gemma 3 4B base and Llama 3.1 8B base. Results are summarized in Table 4.

Table 4: PRR ($\uparrow$ is better) under different sampling strategies. Columns list methods (Dissimilarity, Eccentricity, EigVecDissimilarity) and four different model-dataset pairs; rows list strategies with their hyperparameters. Per column, top-1 is **bold**, second-best is underlined.

| | | Gemma 3 4B base | | | | | | Llama 3.1 8B base | | | | | |
| | | TriviaQA | | | CoQA | | | TriviaQA | | | CoQA | | |
| | | Dissim | Ecc | EigVec Dissim | Dissim | Ecc | EigVec Dissim | Dissim | Ecc | EigVec Dissim | Dissim | Ecc | EigVec Dissim |
|---|---|---|---|---|---|---|---|---|---|---|---|---|---|
| Beam search | | .771 | .732 | .751 | **.561** | .483 | .488 | .623 | .581 | .598 | .502 | **.438** | **.454** |
| Multinomial Sampling | $T = 0.7$ | .703 | .619 | .687 | .455 | .368 | .397 | .561 | .521 | .538 | .451 | .371 | .382 |
| | $T = 0.9$ | .734 | .664 | .710 | .468 | .417 | .425 | .588 | .521 | .533 | .418 | .407 | .410 |
| | $T = 1.0$ | .742 | .689 | .715 | .465 | .424 | .432 | .599 | .516 | .537 | .431 | .416 | .424 |
| | $T = 1.2$ | .733 | .679 | .717 | .435 | .424 | .437 | .610 | .517 | .535 | .391 | .427 | .433 |
| | $T = 1.5$ | .718 | .685 | .717 | .406 | .426 | .436 | .555 | .515 | .532 | .397 | .431 | .440 |
| | $T = 1.7$ | .680 | .690 | .717 | .372 | .426 | .433 | .569 | .514 | .530 | .304 | .432 | .441 |
| Diverse Beam Search | $G = 2, \lambda = 0.5$ | .753 | .528 | .693 | .498 | .405 | .452 | .623 | -.123 | .546 | .458 | .310 | .363 |
| | $G = 2, \lambda = 1.0$ | .753 | .566 | .705 | .518 | .384 | .432 | .594 | -.138 | .539 | .466 | .285 | .355 |
| | $G = 2, \lambda = 1.5$ | .763 | .522 | .714 | .537 | .420 | .441 | .618 | -.101 | .542 | .462 | .245 | .317 |
| | $G = 2, \lambda = 2.0$ | .759 | .515 | .702 | .547 | .377 | .391 | .630 | -.130 | .546 | .452 | .215 | .287 |
| | $G = 5, \lambda = 0.5$ | .758 | .546 | .736 | .493 | .401 | .453 | .591 | -.026 | .569 | .395 | .262 | .353 |
| | $G = 5, \lambda = 1.0$ | .768 | .523 | .746 | .515 | .369 | .423 | .615 | -.086 | .563 | .447 | .274 | .376 |
| | $G = 5, \lambda = 1.5$ | .761 | .453 | .723 | .513 | .391 | .427 | .623 | -.153 | .533 | .461 | .199 | .324 |
| | $G = 5, \lambda = 2.0$ | .770 | .476 | .690 | .513 | .355 | .415 | .631 | -.093 | .548 | .453 | .132 | .254 |
| Hybrid Multinomial-Beam | $B = 1$ | .759 | .715 | .746 | .512 | .451 | .433 | .597 | .519 | .549 | .386 | .314 | .325 |
| | $B = 2$ | .765 | .731 | .745 | .519 | .470 | .435 | .617 | .564 | .586 | .445 | .338 | .345 |
| | $B = 3$ | .781 | .736 | .754 | .503 | .461 | .439 | .620 | .553 | .598 | **.519** | .436 | .424 |
| | $B = 4$ | **.784** | **.750** | **.769** | .516 | .467 | .428 | .622 | .572 | .617 | .436 | .388 | .382 |
| | $B = 5$ | .757 | .733 | .749 | .538 | .500 | .466 | **.655** | .578 | .609 | .470 | .412 | .419 |
| | $B = 6$ | .773 | .733 | .756 | .528 | **.512** | .488 | .635 | .586 | .613 | .486 | .421 | .415 |
| | $B = 7$ | .771 | .737 | .754 | .543 | .468 | .471 | .640 | .596 | .617 | .483 | .399 | .427 |
| | $B = 8$ | .764 | .733 | .755 | .548 | .504 | **.507** | .648 | .597 | .610 | .491 | .434 | .425 |
| | $B = 9$ | .772 | .747 | .765 | .551 | .509 | .497 | .646 | **.597** | **.618** | .501 | .427 | .439 |

No single strategy dominates across datasets, models, or estimators. Temperature sampling (with importance weights) and diverse beam search systematically yield to beam search and hybrid multinomial-beam. Hybrid multinomial–beam strategy can reach top-1 for specific hyperparameter $B$, but gains are not systematic and are sensitive to tuning. Given this variability and tuning cost, plain beam search with probability weighting is a reasonable default.

## A.2 RESTRICTED-MASS NORMALIZATION

Equation (4) normalizes autoregressive sequence probabilities over the $M$ beam candidates. This choice can be sensitive to tail candidates whose probabilities are tiny and length-dependent. To test robustness, we introduce a floor $\epsilon$ on the per-candidate mass:

$$w_i^\epsilon = \frac{\max\big(\epsilon,\, p(\mathbf{b}^{(i)} \mid \mathbf{x})\big)}{\sum_{j=1}^{M} \max\big(\epsilon,\, p(\mathbf{b}^{(j)} \mid \mathbf{x})\big)}. \tag{21}$$

The setting $\epsilon = 0$ recovers equation (4); $\epsilon = 1$ yields uniform weights $w_i^1 = 1/M$. Intermediate $\epsilon$ values trade off fidelity to the model distribution against robustness to noisy, length-biased tails.

We evaluate beam-guided probability-weighted methods for different $\epsilon$ on a subset of 500 examples from TriviaQA and CoQA with two base models, Gemma 3 4B base and Llama 3.1 8B base. Results are summarized in Table 5.

The results do not indicate a clear best choice of method and corresponding $\epsilon$ parameter. Determining the optimal $\epsilon$ is a case-dependent task.

Table 5: PRR ($\uparrow$ is better) under restricted-mass normalization ablation. Columns group dataset–model pairs with methods (Dissim, Ecc, EigVecDissim). Rows vary the mass floor $\epsilon$ in equation (21): $\epsilon = 0$ recovers equation (4); $\epsilon = 1$ yields uniform weights $w_i = 1/M$. For each dataset–method, the top-1 score is **bold** and the second-best is underlined.

| | Gemma 3 4B base | | | | | | Llama 3.1 8B base | | | | | |
| | TriviaQA | | | CoQA | | | TriviaQA | | | CoQA | | |
| | Dissim | Ecc | EigVec Dissim | Dissim | Ecc | EigVec Dissim | Dissim | Ecc | EigVec Dissim | Dissim | Ecc | EigVec Dissim |
|---|---|---|---|---|---|---|---|---|---|---|---|---|
| $\epsilon = 1.0$ | .765 | **.741** | .744 | .536 | .487 | .461 | **.668** | .596 | .607 | .470 | .428 | .410 |
| $\epsilon = 0.1$ | .765 | .727 | .745 | .556 | **.497** | .483 | .667 | **.612** | **.627** | .502 | **.447** | .446 |
| $\epsilon = 0.05$ | .764 | .720 | .744 | .561 | .490 | .487 | .657 | .606 | .626 | **.509** | .435 | .451 |
| $\epsilon = 0.01$ | .766 | .718 | .749 | .559 | .478 | **.489** | .630 | .584 | .602 | .496 | .437 | .452 |
| $\epsilon = 0.001$ | .771 | .731 | **.751** | **.562** | .484 | .488 | .624 | .581 | .598 | .501 | .438 | .453 |
| $\epsilon = 0.00001$ | **.771** | .732 | .751 | .561 | .483 | .488 | .623 | .581 | .598 | .502 | .438 | **.454** |
| $\epsilon = 0$ | **.771** | .732 | .751 | .561 | .483 | .488 | .623 | .581 | .598 | .502 | .438 | .454 |

## A.3 OTHER SAMPLING-BASED METHODS UNDER BEAM SEARCH

Beyond Dissimilarity, Eccentricity, and EigVecDissimilarity, this ablation evaluates two other sampling-based methods under beam-generated candidates: *Degree Matrix* (Lin et al., 2024) and *Semantic Entropy* (Kuhn et al., 2023). We also provide probability-weighted beam formulations using the weights $w_i$ from equation (4).

**Degree Matrix.** Given $M$ multinomial samples $\{\mathbf{y}^{(i)}\}_{i=1}^{M}$, Degree Matrix estimates the average pairwise dissimilarity:

$$\widehat{U}_{DegMat}(\mathbf{x}) = \frac{1}{M^2} \sum_{i=1}^{M} \sum_{j=1}^{M} \big(1 - s(\mathbf{y}^{(i)}, \mathbf{y}^{(j)})\big). \tag{22}$$

For beam candidates $\{\mathbf{b}^{(i)}\}_{i=1}^{M}$, our mass-aware variant averages with weights:

$$\widehat{U}_{DegMat}^{b}(\mathbf{x}) = \sum_{i=1}^{M} w_i \sum_{j=1}^{M} w_j \big(1 - s(\mathbf{b}^{(i)}, \mathbf{b}^{(j)})\big). \tag{23}$$

Table 6: PR-AUC (↑ is better) on 6 datasets with Gemma 3 4B base. Each method is shown as a pair: its multinomial-sampling variant and its beam-search variant; ↑ denotes an improvement of the beam variant over its multinomial counterpart. Along main methods, the table includes input-uncertainty methods (Semantic Entropy, Lexical Similarity). For each dataset, the top-1 score is **bold** and the second-best is underlined. The rightmost column reports the mean PR-AUC across datasets.

| UQ Method | TriviaQA | Web Questions | CoQA | HotpotQA | Common senceQA | ARC-Challenge | Mean |
|---|---|---|---|---|---|---|---|
| Semantic Entropy | .622 | .505 | .301 | .140 | .407 | .431 | .401 |
| Semantic Entropy + beamsearch | .685↑ | .614↑ | .365↑ | .278↑ | .436↑ | .454↑ | .472↑ |
| Degree Matrix | .682 | .605 | .385 | .311 | .409 | .419 | .469 |
| Degree Matrix + beamsearch | .673 | .642↑ | .328 | .244 | .444↑ | .473↑ | .467 |
| SAR | .656 | .571 | .347 | .296 | .183 | .264 | .386 |
| SAR + beamsearch | .671↑ | .589↑ | .329 | .266 | .209↑ | .269↑ | .372 |
| Dissimilarity | .755 | .715 | .578 | .626 | .561 | .545 | .630 |
| Dissimilarity + beamsearch | **.766**↑ | **.722**↑ | **.600**↑ | .611 | **.595**↑ | .604↑ | **.650**↑ |
| Eccentricity | .714 | .653 | .459 | .453 | .549 | .549 | .563 |
| Eccentricity + beamsearch | .739↑ | .633 | .505↑ | .514↑ | .590↑ | **.636**↑ | .603↑ |
| EigVecDissimilarity | .738 | .661 | .443 | .448 | .512 | .562 | .561 |
| EigVecDissimilarity + beamsearch | .753↑ | .668↑ | .497↑ | .487↑ | .562↑ | .621↑ | .598↑ |
| CocoaMSP | .738 | .666 | .509 | .430 | .583 | .595 | .587 |
| CocoaMSP + beamsearch | .747↑ | .679↑ | .548↑ | .523↑ | .586↑ | .606↑ | .615↑ |
| CocoaPPL | .739 | .678 | .548 | .625 | .580 | .595 | .628 |
| CocoaPPL + beamsearch | .748↑ | .694↑ | .577↑ | **.681**↑ | .582↑ | .610↑ | .649↑ |

**Semantic Entropy.** Multinomial samples are clustered into semantic equivalence classes $C$. For each class, we calculate its probability

$$\hat{p}(c) = \frac{1}{M}\sum_{i=1}^{M}\mathbf{1}\{\mathbf{y}^{(i)} \in c\} \quad \text{for } c \in C. \tag{24}$$

Then Semantic Entropy calculates

$$\widehat{U}_{SemEnt}(\mathbf{x}) = -\frac{1}{|C|}\sum_{c \in C}\log\hat{p}(c). \tag{25}$$

For beam candidates, use cluster masses aggregated by $w_i$:

$$\hat{p}^b(c) = \sum_{i=1}^{M}w_i\,\mathbf{1}\{\mathbf{b}^{(i)} \in c\}, \qquad \widehat{U}^b_{SemEnt}(\mathbf{x}) = -\sum_{c \in C}\hat{p}^b(c)\log\hat{p}^b(c). \tag{26}$$

Note that these objectives score LLM uncertainty about the *input* $\mathbf{x}$ as they are independent of a particular $\mathbf{y}_*$.

The results are summarized in Table 6. Beam search yields significant gains for Semantic Entropy and little to no improvement for Degree Matrix. Even with the beam-adapted formulations above, both objectives show worse results in terms of absolute PR-AUC compared to other methods. The primary reason is the target mismatch: as noted, these scores quantify uncertainty of the input $\mathbf{x}$ and are independent of the produced answer $\mathbf{y}_*$, whereas our main methods, Dissimilarity, Eccentricity and EigVecDissimilarity, focuses on ranking the correctness of $\mathbf{y}_*$ itself.

To further assess performance under different numbers of samples $M$ used for UQ, we plot PRR as a function of $M$ for one selected baseline, Semantic Entropy, as well as for Dissimilarity (both sampling and beam-search variants) for reference. Figure 7 presents the results, showing that for all $M > 1$, Semantic Entropy underperforms both Dissimilarity variants. This occurs because Dissimilarity measures the targeted uncertainty of the specific generation $\mathbf{y}_*$ rather than the overall uncertainty associated with $\mathbf{x}$, measured by Semantic Entropy.

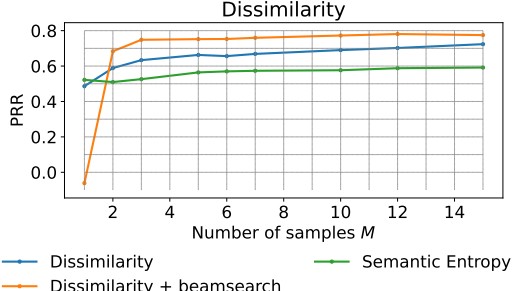

Figure 7: PRR ($\uparrow$ is better) as a function of the number of candidates $M$ on TriviaQA with Gemma 3 4B base for 3 UQ methods: Semantic Entropy, and sampling and beam search versions of Dissimilarity.

Table 7: PRR ($\uparrow$ is better) for Eccentricity and EigVecDissimilarity under different Graph Laplacian embedding choices on four dataset–model pairs. Top block varies the eigenvalue threshold $\alpha$ (retaining all $\lambda_i < \alpha$); bottom block fixes the embedding dimension $K$. For each pair, the best score is **bold** and the second-best is underlined.

| | Gemma 3 4B base | | | | Llama 3.1 8B base | | | |
| | TriviaQA | | CoQA | | TriviaQA | | CoQA | |
| | Ecc | EigVec Dissim | Ecc | EigVec Dissim | Ecc | EigVec Dissim | Ecc | EigVec Dissim |
|---|---|---|---|---|---|---|---|---|
| $\alpha = 0.3$ | .717 | .710 | .434 | .355 | .601 | .599 | .408 | .364 |
| $\alpha = 0.5$ | **.752** | .740 | .497 | .460 | **.627** | .626 | .431 | .420 |
| $\alpha = 0.7$ | .750 | .749 | **.539** | **.498** | .622 | **.643** | .438 | .441 |
| $\alpha = 0.8$ | .751 | **.757** | .508 | .475 | .616 | .630 | **.444** | .450 |
| $\alpha = 0.9$ | .732 | .751 | .483 | .488 | .581 | .598 | .438 | **.454** |
| $\alpha = 0.99$ | .725 | .755 | .432 | .454 | .535 | .561 | .397 | .409 |
| $K = 1$ | .454 | .358 | .346 | .332 | .444 | .357 | .304 | .280 |
| $K = 2$ | .510 | .532 | .383 | .361 | .474 | .469 | .318 | .338 |
| $K = 3$ | .619 | .639 | .434 | .429 | .538 | .534 | .351 | .348 |
| $K = 4$ | .645 | .643 | .418 | .412 | .519 | .514 | .359 | .352 |
| $K = 5$ | .638 | .655 | .366 | .352 | .487 | .492 | .314 | .316 |
| $K = 6$ | .529 | .545 | .244 | .236 | .363 | .368 | .210 | .211 |
| $K = 7$ | .210 | .242 | -.101 | -.076 | .050 | .062 | -.208 | -.211 |
| $K = 8$ | -.265 | -.209 | -.210 | -.171 | -.358 | -.349 | -.327 | -.308 |
| $K = 9$ | -.566 | -.414 | -.268 | -.186 | -.462 | -.410 | -.339 | -.317 |
| $K = 10$ | -.659 | -.484 | -.283 | -.189 | -.467 | -.397 | -.330 | -.249 |

## A.4 GRAPH LAPLACIAN EMBEDDING PARAMETERS

Both multinomial and beam-guided versions of Eccentricity and EigVecDissimilarity depend on the threshold parameter $\alpha$, which selects eigenvectors of the Graph Laplacian $L = I - D^{-1/2}WD^{-1/2}$ used to form semantic embeddings. Specifically, after computing the eigenpairs $\{\lambda_i, \mathbf{u}_i\}_{i=1}^{M+1}$, we retain those with $\lambda_i < \alpha$, yielding $K$ eigenvectors in total and embeddings $\mathbf{v}_j = [\mathbf{u}_{1j}, \mathbf{u}_{2j}, \ldots, \mathbf{u}_{Kj}]$ of dimension $K$. All eigenvalues lie in $[0, 1]$; smaller values capture stronger semantic structure, whereas values closer to 1 tend to reflect noise (Lin et al., 2024). In the main experiments we follow the original Eccentricity setting and use $\alpha = 0.9$.

Here we vary $\alpha$ and also test a fixed-$K$ strategy (i.e., keeping exactly $K$ leading low-spectrum eigenvectors irrespective of the threshold).

Table 7 reports the performance for Eccentricity and EigVecDissimilarity across $\alpha$ and $K$ on four dataset–model pairs. A fixed embedding size performs poorly: the optimal number of informative directions varies between candidate sets, so fixing $K$ either underfits or includes noisy directions. Thresholding is more robust: $\alpha \in [0.7, 0.9]$ consistently yields strong results across methods and pairs, supporting our default choice $\alpha = 0.9$.

## A.5 CROSS-ENCODER SIMILARITY

In the main text, we instantiate the similarity function $s$ using an NLI score: the entailment probability from a DeBERTa model. CoCoA, however, originally used a RoBERTa-large *cross-encoder* fine-tuned on the Semantic Textual Similarity benchmark (Liu et al., 2019). Table 8 reports PRR for Gemma 3 4B base when replacing the NLI-based $s$ with this cross-encoder; all other settings are unchanged.

Table 8: PRR (↑ is better) on 6 datasets with Gemma 3 4B base using a RoBERTa-large cross-encoder (STS) as the similarity function $s$ in place of NLI. For each dataset, the top-1 is **bold** and the second-best is underlined; ↑ marks an improvement of a beam variant over its multinomial counterpart.

| UQ Method | TriviaQA | Web Questions | CoQA | HotpotQA | Common senceQA | ARC-Challenge | Mean |
|---|---|---|---|---|---|---|---|
| Dissimilarity | .725 | .683 | .497 | .597 | .481 | .421 | .567 |
| Dissimilarity + beamsearch | **.746**↑ | **.693**↑ | **.513**↑ | **.654**↑ | .505↑ | .479↑ | .598↑ |
| Eccentricity | .722 | .647 | .489 | .544 | .455 | .500 | .560 |
| Eccentricity + beamsearch | .734↑ | .647↑ | .483 | .604↑ | .362 | .421 | .542 |
| EigVecDissimilarity | .737 | .649 | .453 | .523 | .489 | .529 | .563 |
| EigVecDissimilarity + beamsearch | .744↑ | .675↑ | .484↑ | .582↑ | .439 | .496 | .570↑ |
| CocoaMSP | .731 | .642 | .438 | .397 | .553 | .577 | .556 |
| CocoaMSP + beamsearch | .740↑ | .648↑ | .462↑ | .479↑ | **.558**↑ | **.593**↑ | .580↑ |
| CocoaPPL | .728 | .653 | .488 | .607 | .546 | .567 | .598 |
| CocoaPPL + beamsearch | .737↑ | .658↑ | .498↑ | .650↑ | .548↑ | .586↑ | **.613**↑ |

## A.6 NUMBER OF SAMPLES ACROSS DIFFERENT TASKS

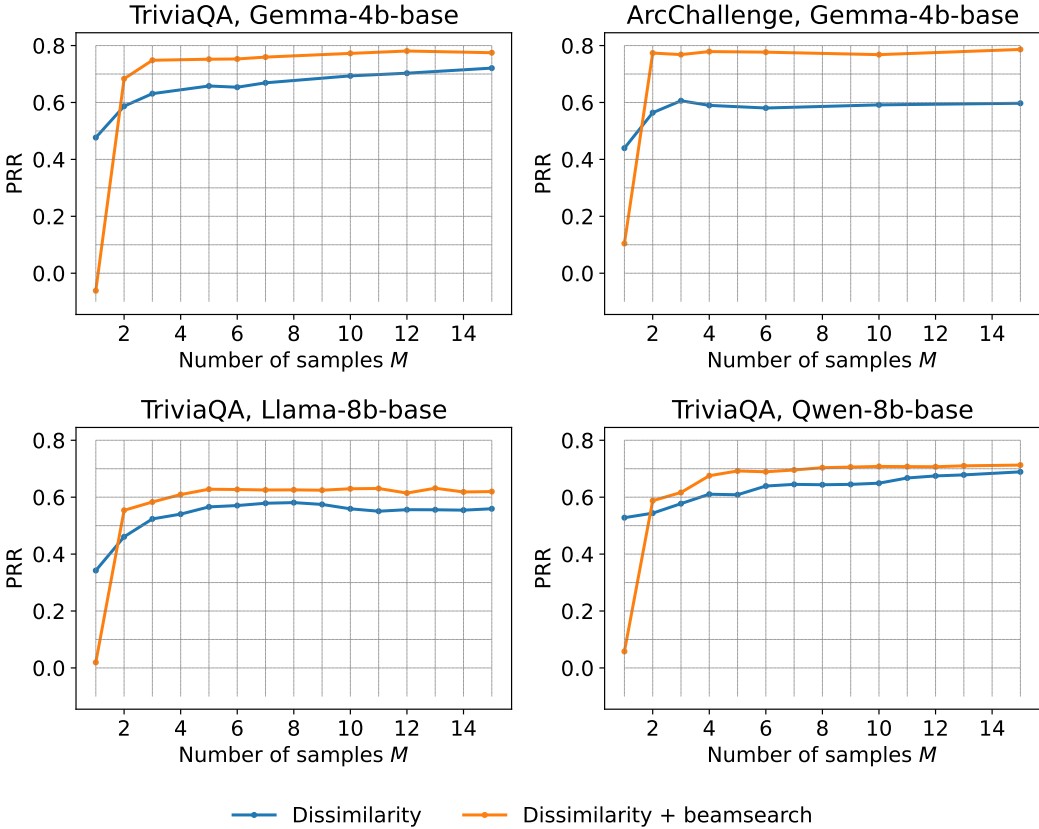

Figure 8: PRR (↑ is better) as a function of the number of candidates $M$ across different datasets and models.

To evaluate the performance of the proposed beam-search variations under different numbers of samples $M$ across models, we computed PRR for both the sampling and beam-search versions of Dissimilarity on 200 random subsamples of TriviaQA for three LLMs: Gemma 3 4B base, Llama 3.1 8B base, and Qwen 3 8B base. To further assess performance across datasets, we additionally evaluated PRR for Gemma 3 4B base on the 200 subsamples of ARC-Challenge dataset. All four resulting plots are shown in Figure 8.

The results show that for all budgets $M > 1$, beam search consistently outperforms sampling, yielding higher PRR. On the open-ended TriviaQA dataset, PRR increases steadily with $M$, with beam search reaching a plateau around $M = 5$ for all 3 models tested. On the multiple-choice ARC-Challenge dataset, PRR plateaus at a considerably smaller budget ($M = 2$), likely due to the small output space (i.e., a limited set of answer choices).

Overall, these results indicate that the beam-search variant of Dissimilarity remains effective even at relatively small sample budgets: $M \approx 5$ for open-ended short-form generation tasks, and $M = 2$ for multiple-choice settings, where the constrained output space enables faster saturation.

# B   ANALYSIS AND EXAMPLES

## B.1   PROBABILITY MASS COVERAGE

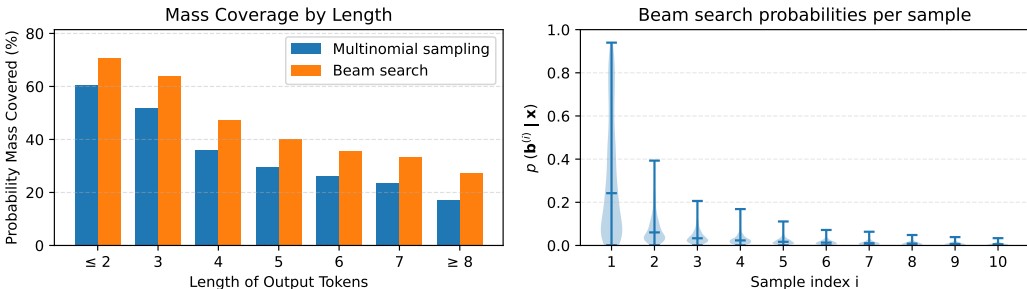

Figure 9: *Left*: average probability mass covered by the candidate set ($M{=}10$) across output-length bins (averaged over examples in the bin) on TriviaQA with Gemma 3 4B base. *Right*: for beam search, distribution of sequence probabilities $p(\mathbf{b}^{(i)} \mid \mathbf{x})$ by beam rank $i$ (1 = highest-probability text).

Figure 9 summarizes two observations. First (left), beam search covers a larger share of the model's probability mass than multinomial sampling across length bins. Second (right), beam probabilities decay sharply with rank: the first few beams capture most of the mass, while lower-ranked beams contribute little. This motivates mass-aware weighting $w_i$ (see equation (4)) and helps explain why probability-weighted beam variants are effective, especially at small candidate budgets.

## B.2   EXAMPLES

We include qualitative examples for Gemma 3 4B base: two from TriviaQA, two from WebQuestions, and one from CoQA. Each panel shows the question, the greedy answer, ten multinomial samples, and ten beam-search samples with autoregressive probabilities, together with the corresponding uncertainty scores (e.g., Dissimilarity and its beam-guided variant). The cases illustrate how beam search reduces duplication and enhances uncertainty.

**Question:** What claimed the life of singer Kathleen Ferrier?
**Greedy:** breasts cancer

| Multinomial samples | Beam-search samples | |
| --- | --- | --- |
| cancer | cancer | p=0.228 |
| breast cancer | tuberculosis | p=0.154 |
| pulmonary | breast cancer | p=0.089 |
| breast cancer | lung cancer | p=0.041 |
| cancer | pneumonia | p=0.039 |
| breast cancer | leukaemia | p=0.034 |
| myx | myel | p=0.023 |
| cancer | leuk | p=0.011 |
| cancer | pulmonary | p=0.011 |
| pneumonia | lymphoma | p=0.010 |

**Dissimilarity:** 0.330
**Dissimilarity + beamsearch:** 0.533

**Question:** Which number Beethoven symphony is known as 'The Pastoral'?
**Greedy:** 6

| Multinomial samples | Beam-search samples | |
| --- | --- | --- |
| six | sixth | p=0.314 |
| seventh | 6 | p=0.169 |
| sixth | 6th | p=0.104 |
| sixth | ninth | p=0.061 |
| sixth | seventh | p=0.037 |
| 6 | 9 | p=0.027 |
| seventh | six | p=0.023 |
| no | 9th | p=0.021 |
| sixteenth | no. | p=0.013 |
| n6 | 7 | p=0.008 |

**Dissimilarity:** 0.634
**Dissimilarity + beamsearch:** 0.561

Figure 10: Two examples from Gemma 3 4B base on TriviaQA. Each panel shows the question, greedy answer, multinomial and beam-search samples with autoregressive probabilities, plus dissimilarity and beamsearch-guided dissimilarity.

**Question:** what currency does cyprus use?
**Greedy:** Cyprus pound

| Multinomial samples | Beam-search samples | |
|---|---|---|
| Euro | Euro | p=0.439 |
| Euro | euro | p=0.201 |
| Euro | Cyprus pound | p=0.091 |
| Euro | Cypriot | p=0.072 |
| euro | Cyprus Pound | p=0.016 |
| euro | Euros | p=0.014 |
| euro | EURO | p=0.007 |
| euros | euros | p=0.007 |
| Euro | cyprus | p=0.007 |
| Euro | Cyprus | p=0.006 |

**Dissimilarity:** 0.976
**Dissimilarity + beamsearch:** 0.800

**Question:** who plays charlie in the santa clause movies?
**Greedy:** Tim Allen

| Multinomial samples | Beam-search samples | |
|---|---|---|
| Tim Allen | Tim Allen | p=0.318 |
| Tim Allen | Jeff Daniels | p=0.017 |
| Scott Calvin | Timothy Oly | p=0.012 |
| Tim Allen | Ed Asner | p=0.010 |
| Tim Allen | Scott Calvin | p=0.008 |
| Edward Arnold | Edward Asner | p=0.008 |
| Tim Allen | Tony Cox | p=0.007 |
| Jeremy nault | Tim Allen | p=0.007 |
| Tim Allen | Tim allen | p=0.005 |
| Tim Allen | Eric Lloyd | p=0.004 |

**Dissimilarity:** 0.427
**Dissimilarity + beamsearch:** 0.313

Figure 11: Two examples from Gemma 3 4B base on WebQ. Each panel shows the question, greedy answer, multinomial and beam-search samples with autoregressive probabilities, plus dissimilarity and beamsearch-guided dissimilarity.

**Story:** A couple of weeks ago, my 12-year-old daughter, Ella threatened to take my phone and break it. "At night you'll always have your phone out and break you'll just type," Ella says. "I'm ready to go to bed, and try to get you to read stories for me and you're just standing there reading your texts and texting other people," she adds. I came to realize that I was ignoring her as a father...

**Question:** She mentions a lot of grown ups don't make what in their lifetime?
**Greedy:** Limits.

| Multinomial samples | Beam-search samples | |
|---|---|---|
| Boundaries. | Boundaries. | p=0.185 |
| Set limits. | Limits. | p=0.079 |
| Boundaries. | They don't | p=0.040 |
| limits. | Rules. | p=0.032 |
| Boundaries in their | Boundaries. | p=0.029 |
| Charging station. | Boundaries. | p=0.028 |
| Similar limitations. | A charging station. | p=0.016 |
| Boundaries. | Boundaries that protect | p=0.010 |
| Boundaries. | Limits in their own | p=0.007 |
| Set up similar limits | Boundaries | p=0.007 |

**Dissimilarity:** 0.310
**Dissimilarity + beamsearch:** 0.190

Figure 12: One example from Gemma 3 4B base on CoQA. Shown are the question, greedy answer, multinomial and beam-search samples with autoregressive probabilities, plus dissimilarity and beamsearch-guided dissimilarity.

## C  DATASETS

Table 9 lists the prompts used to form inputs for each dataset (separately for base and instruct models). Table 10 reports mean accuracy for each model–dataset pair. We measure accuracy as the fraction of predictions whose AlignScore with the gold answer exceeds 0.5.

Table 9: Prompt templates used for each dataset and model type. Few-shot exemplars are shown as placeholders (e.g., `<5 few-shot QA pairs>`); run-time inputs are denoted by `<question>`, `<context>`, `<title 1>`, etc.

| Dataset | Base Prompt | Instruct Prompt |
|---|---|---|
| TriviaQA | `<5 few-shot QA pairs>`
`Question: <question>`
`Answer:` | `Answer the following question as briefly as possible.`
`<5 few-shot QA pairs>`
`Now answer the following question:`
`Question: <question>`
`Answer:` |
| Web Questions | `<5 few-shot QA pairs>`
`Question: <question>`
`Answer:` | `Below are questions with short factual answers.`
`Return only the short answer (a name, phrase, number,`
`or year).`
`<5 few-shot QA pairs>`
`Now answer this.`
`Q: <question>`
`A:` |
| CoQA | `Story: <context>`
`<all preceding QA pairs>`
`Question: <question>`
`Answer:` | `Story: <context>`
`<all preceding QA pairs>`
`Answer the following question as briefly as possible.`
`Question: <question>`
`Answer:` |
| HotpotQA | `Title: <title 1>`
`<paragraph 1>`
`Title: <title 2>`
`<paragraph 2>`
`Question: <question>`
`Short answer:` | `Instruction: Read the context and answer with a`
`short factual span (a few words) copied from the`
`context. Reply with the short answer only.`
`Title: <title 1>`
`<paragraph 1>`
`Title: <title 2>`
`<paragraph 2>`
`Question: <question>`
`Short answer:` |
| Common senceQA | `<2 few-shot QA pairs>`
`Question: <question>`
`Options: <(A) - (D)`
`options>`
`Answer:` | `Instruction: Choose the single best answer from the`
`options. Answer with the option text only (not the`
`letter).`
`<2 few-shot QA pairs>`
`Now answer this.`
`Question: <question>`
`Options:`
`<(A) - (D) options>`
`Answer:` |
| ARC-Challenge | `<2 few-shot QA pairs>`
`Question: <question>`
`Options:`
`<(A) - (D) options>`
`Answer:` | `Instruction: Choose the single best answer from the`
`options. Answer with the option text only (not the`
`letter).`
`<2 few-shot QA pairs>`
`Now answer this.`
`Question: <question>`
`Options:`
`<(A) - (D) options>`
`Answer:` |

Table 10: Mean accuracy (%): proportion of predictions with AlignScore to the gold answer $> 0.5$.

| | Closed-Book QA | | Open-Book QA | | Multiple Choice | |
|---|---|---|---|---|---|---|
| | TriviaQA | Web Questions | CoQA | HotpotQA | Common senceQA | ARC-Challenge |
| Llama 3.1 8B base | 63% | 47% | 74% | 53% | 74% | 72% |
| Llama 3.1 8B instruct | 69% | 40% | 80% | 72% | 77% | 76% |
| Gemma 3 4B base | 47% | 33% | 69% | 41% | 65% | 70% |
| Gemma 3 4B instruct | 51% | 35% | 76% | 66% | 76% | 77% |
| Qwen 3 8B base | 52% | 48% | 81% | 47% | 89% | 91% |
| Qwen 3 8B instruct | 54% | 42% | 76% | 76% | 84% | 88% |

# D  ADDITIONAL RESULTS

## D.1  SCORING TOP-BEAM OUTPUT

In the main text we score the greedy decode as the produced answer $\mathbf{y}_*$. Table 11 complements these results by scoring the *top-1 beam* as $\mathbf{y}_*$, a natural choice when beam search is already used to obtain a higher-quality decode. The beam-weighted family of approaches achieves higher PRR than the original methods and baselines in the majority of cases.

Table 11: PRR ($\uparrow$ is better) averaged over 6 datasets, when scoring the top-1 beam produced answer (instead of greedy). For each dataset, the top-1 score is **bold** and the second-best is underlined. For beam-guided variants, we mark $\uparrow$ when the variant improves over its original multinomial-sampling counterpart.

| UQ Method | Llama 3.1 8B | | Gemma 3 4B | | Qwen 3 8B | |
|---|---|---|---|---|---|---|
| | base | instruct | base | instruct | base | instruct |
| *Baseline UQ methods* | | | | | | |
| Prob | .399 | .174 | .400 | .213 | .390 | .090 |
| MTE | .320 | .164 | .317 | .228 | .334 | .255 |
| Perplexity | .376 | .121 | .359 | .185 | .318 | .009 |
| CCP | .395 | .155 | .369 | .243 | .352 | .226 |
| SAR | .333 | .221 | .336 | .348 | .342 | .246 |
| P(True) | .019 | -.075 | .031 | .090 | .012 | -.080 |
| SemanticEntropy | .345 | .286 | .397 | .320 | .299 | .250 |
| LexicalSimilarity | .377 | .221 | .384 | .291 | .404 | .210 |
| EigValLaplacian | .366 | .209 | .402 | .307 | .384 | .223 |
| NumSemSets | .349 | .215 | .365 | .262 | .344 | .208 |
| *Consistency-based UQ: multinomial vs. beamsearch versions* | | | | | | |
| Dissimilarity | .437 | .229 | .424 | .333 | .446 | .272 |
| Dissimilarity + beamsearch | .455↑ | .266↑ | .466↑ | .390↑ | .440 | **.346**↑ |
| Eccentricity | .405 | .238 | .395 | .310 | .375 | .208 |
| Eccentricity + beamsearch | .444↑ | .301↑ | .450↑ | .348↑ | .380↑ | .308↑ |
| EigVecDissimilarity | .402 | .243 | .412 | .316 | .403 | .213 |
| EigVecDissimilarity + beamsearch | .446↑ | **.316**↑ | .457↑ | .366↑ | .415↑ | .334↑ |
| CocoaMSP | .447 | .284 | .450 | .347 | .454 | .272 |
| CocoaMSP + beamsearch | **.471**↑ | .290↑ | **.478**↑ | **.407**↑ | **.459**↑ | .345↑ |
| CocoaPPL | .440 | .251 | .433 | .340 | .422 | .261 |
| CocoaPPL + beamsearch | .450↑ | .273↑ | .444↑ | .395↑ | .410 | .318↑ |

## D.2  ROC-AUC AND PR-AUC

In the main text we report PRR. Tables 12 and 13 complements these results with ROC-AUC and PR-AUC on Gemma 3 4B base. We binarize by marking an answer as correct if its AlignScore to the gold answer exceeds $0.5$, and incorrect otherwise (the positive class for PR-AUC is the incorrect label). The pattern mirrors PRR: beam-guided variants generally match or outperform multinomial sampling.

## D.3  DETAILED RESULTS FOR EACH DATASET

Complementing the main-table results in Table 3, Tables 14–19 report PRR for six datasets separately for Gemma 3 4B base, Gemma 3 4B instruct, Llama 3.1 8B base, Llama 3.1 8B instruct, Qwen 3 8B base, and Qwen 3 8B instruct.

Table 12: ROC-AUC↑ for 6 datasets with Gemma 3 4B base. For each dataset, the top-1 method is **bold** and the second-best is underlined. Beam-guided and probability-weighted variants are marked with ↑ when they improve over their multinomial-sampling baseline. The two rightmost columns report the mean ROC-AUC across datasets.

| UQ Method | TriviaQA | Web Questions | CoQA | HotpotQA | Common senceQA | ARC-Challenge | Mean |
|---|---|---|---|---|---|---|---|
| *Baseline UQ methods* | | | | | | | |
| Prob | .863 | .768 | .698 | .632 | .796 | .821 | .763 |
| MTE | .867 | .793 | .710 | .721 | .737 | .753 | .763 |
| Perplexity | .863 | .785 | .729 | .735 | .796 | .820 | .788 |
| CCP | .881 | .781 | .698 | .660 | .775 | .793 | .764 |
| SAR | .867 | .776 | .701 | .713 | .653 | .696 | .748 |
| P(True) | .642 | .473 | .524 | .513 | .571 | .545 | .545 |
| SemanticEntropy | .849 | .758 | .690 | .591 | .755 | .774 | .736 |
| LexicalSimilarity | .842 | .766 | .713 | .656 | .739 | .756 | .745 |
| EigValLaplacian | .867 | .766 | .701 | .633 | .739 | .775 | .747 |
| NumSemSets | .856 | .754 | .653 | .639 | .702 | .757 | .727 |
| *Consistency-based UQ: multinomial vs. beamsearch versions* | | | | | | | |
| Dissimilarity | .916 | .836 | .822 | .809 | .817 | .818 | .836 |
| Dissimilarity + beamsearch | **.923**↑ | **.852**↑ | **.826**↑ | .814↑ | .831↑ | .841↑ | **.848**↑ |
| Eccentricity | .897 | .808 | .768 | .737 | .809 | .821 | .806 |
| Eccentricity + beamsearch | .911↑ | .816↑ | .790↑ | .771↑ | **.833**↑ | **.859**↑ | .830↑ |
| EigVecDissimilarity | .902 | .813 | .761 | .728 | .798 | .825 | .805 |
| EigVecDissimilarity + beamsearch | .920↑ | .827↑ | .787↑ | .763↑ | .820↑ | .856↑ | .829↑ |
| CocoaMSP | .904 | .823 | .791 | .726 | .826 | .839 | .818 |
| CocoaMSP + beamsearch | .910↑ | .836↑ | .811↑ | .779↑ | .827↑ | .847↑ | .835↑ |
| CocoaPPL | .907 | .832 | .810 | .799 | .825 | .837 | .835 |
| CocoaPPL + beamsearch | .912↑ | .845↑ | .823↑ | **.828**↑ | .825↑ | .844↑ | .846↑ |

Table 13: PR-AUC↑ for 6 datasets with Gemma 3 4B base. For each dataset, the top-1 method is **bold** and the second-best is underlined. Beam-guided and probability-weighted variants are marked with ↑ when they improve over their multinomial-sampling baseline. The two rightmost columns report the mean PR-AUC across datasets.

| UQ Method | TriviaQA | Web Questions | CoQA | HotpotQA | Common senceQA | ARC-Challenge | Mean |
|---|---|---|---|---|---|---|---|
| *Baseline UQ methods* | | | | | | | |
| Prob | .855 | .838 | .477 | .678 | .623 | .628 | .683 |
| MTE | .875 | .874 | .545 | .799 | .558 | .540 | .699 |
| Perplexity | .860 | .861 | .539 | .814 | .629 | .632 | .722 |
| CCP | .866 | .853 | .475 | .715 | .676 | .641 | .704 |
| SAR | .865 | .861 | .484 | .753 | .437 | .422 | .646 |
| P(True) | .657 | .662 | .326 | .634 | .410 | .355 | .507 |
| SemanticEntropy | .838 | .823 | .456 | .649 | .572 | .511 | .642 |
| LexicalSimilarity | .833 | .848 | .514 | .711 | .545 | .509 | .660 |
| EigValLaplacian | .865 | .855 | .481 | .682 | .565 | .532 | .663 |
| NumSemSets | .841 | .825 | .427 | .685 | .508 | .497 | .631 |
| *Consistency-based UQ: multinomial vs. beamsearch versions* | | | | | | | |
| Dissimilarity | .911 | .904 | **.715** | .838 | .722 | .648 | .789 |
| Dissimilarity + beamsearch | **.919**↑ | **.915**↑ | .660 | .822 | **.754**↑ | .693↑ | .794↑ |
| Eccentricity | .888 | .887 | .561 | .758 | .685 | .625 | .734 |
| Eccentricity + beamsearch | .906↑ | .884 | .576↑ | .789↑ | .744↑ | **.717**↑ | .769↑ |
| EigVecDissimilarity | .902 | .889 | .573 | .766 | .677 | .651 | .743 |
| EigVecDissimilarity + beamsearch | .916↑ | .900↑ | .588↑ | .784↑ | .717↑ | .689↑ | .766↑ |
| CocoaMSP | .897 | .894 | .605 | .761 | .711 | .680 | .758 |
| CocoaMSP + beamsearch | .907↑ | .904↑ | .632↑ | .801↑ | .715↑ | .691↑ | .775↑ |
| CocoaPPL | .902 | .902 | .672 | .861 | .712 | .686 | .789 |
| CocoaPPL + beamsearch | .909↑ | .910↑ | .690↑ | **.881**↑ | .718↑ | .695↑ | **.801**↑ |

Table 14: PRR (↑ is better) for 6 datasets with Gemma 3 4B base. For each dataset, the top-1 method is **bold** and the second-best is underlined. Beam-guided and probability-weighted variants are marked with ↑ when they improve over their multinomial-sampling baseline.

| Method | TriviaQA | Web Questions | CoQA | HotpotQA | Common senceQA | ARC-Challenge |
|---|---|---|---|---|---|---|
| *Baseline UQ methods* | | | | | | |
| Prob | .659 ± 0.018 | .521 ± 0.031 | .312 ± 0.024 | .274 ± 0.014 | .511 ± 0.025 | .548 ± 0.077 |
| MTE | .670 ± 0.013 | .583 ± 0.029 | .363 ± 0.02 | .494 ± 0.034 | .364 ± 0.031 | .381 ± 0.052 |
| Perplexity | .647 ± 0.024 | .553 ± 0.022 | .369 ± 0.02 | .527 ± 0.023 | .503 ± 0.022 | .547 ± 0.062 |
| CCP | .686 ± 0.021 | .569 ± 0.031 | .326 ± 0.022 | .337 ± 0.025 | .506 ± 0.034 | .527 ± 0.062 |
| SAR | .656 ± 0.02 | .571 ± 0.028 | .347 ± 0.023 | .296 ± 0.018 | .183 ± 0.037 | .264 ± 0.055 |
| P(True) | .272 ± 0.026 | -.004 ± 0.034 | .031 ± 0.026 | .075 ± 0.025 | .090 ± 0.028 | .090 ± 0.048 |
| SemanticEntropy | .622 ± 0.021 | .505 ± 0.022 | .301 ± 0.019 | .140 ± 0.022 | .407 ± 0.028 | .431 ± 0.051 |
| Lexical Similarity | .602 ± 0.017 | .540 ± 0.032 | .349 ± 0.025 | .286 ± 0.016 | .386 ± 0.032 | .392 ± 0.054 |
| EigValLaplacian | .666 ± 0.014 | .555 ± 0.028 | .320 ± 0.036 | .246 ± 0.024 | .386 ± 0.027 | .452 ± 0.046 |
| NumSemSets | .656 ± 0.017 | .538 ± 0.028 | .257 ± 0.027 | .268 ± 0.019 | .338 ± 0.03 | .454 ± 0.042 |
| *Consistency-based UQ: multinomial vs. beamsearch versions* | | | | | | |
| Dissimilarity | .755 ± 0.019 | .715 ± 0.03 | .578 ± 0.022 | .626 ± 0.016 | .561 ± 0.04 | .545 ± 0.062 |
| Dissimilarity + beamsearch | **.766** ↑± 0.023 | **.722** ↑± 0.028 | **.600** ↑± 0.016 | .611 ± 0.021 | **.595** ↑± 0.028 | .604 ↑± 0.052 |
| Eccentricity | .714 ± 0.012 | .653 ± 0.029 | .459 ± 0.02 | .453 ± 0.026 | .549 ± 0.034 | .549 ± 0.054 |
| Eccentricity + beamsearch | .739 ↑± 0.019 | .633 ± 0.035 | .505 ↑± 0.025 | .514 ↑± 0.027 | .590 ↑± 0.024 | **.636** ↑± 0.066 |
| EigVecDissimilarity | .738 ± 0.021 | .661 ± 0.027 | .443 ± 0.031 | .448 ± 0.02 | .512 ± 0.032 | .562 ± 0.035 |
| EigVecDissimilarity + beamsearch | .753 ↑± 0.028 | .668 ↑± 0.032 | .497 ↑± 0.021 | .487 ↑± 0.016 | .562 ↑± 0.028 | .621 ↑± 0.06 |
| CocoaMSP | .738 ± 0.023 | .666 ± 0.028 | .509 ± 0.021 | .430 ± 0.028 | .583 ± 0.03 | .595 ± 0.052 |
| CocoaMSP + beamsearch | .747 ↑± 0.02 | .679 ↑± 0.02 | .548 ↑± 0.02 | .523 ↑± 0.027 | .586 ↑± 0.029 | .606 ↑± 0.072 |
| CocoaPPL | .739 ± 0.015 | .678 ± 0.025 | .548 ± 0.019 | .625 ± 0.023 | .580 ± 0.031 | .595 ± 0.039 |
| CocoaPPL + beamsearch | .748 ↑± 0.024 | .694 ↑± 0.024 | .577 ↑± 0.024 | **.681** ↑± 0.019 | .582 ↑± 0.035 | .610 ↑± 0.048 |

Table 15: PRR (↑ is better) for 6 datasets with Gemma 3 4B instruct. For each dataset, the top-1 method is **bold** and the second-best is underlined. Beam-guided and probability-weighted variants are marked with ↑ when they improve over their multinomial-sampling baseline.

| UQ Method | TriviaQA | Web Questions | CoQA | HotpotQA | Common senceQA | ARC-Challenge |
|---|---|---|---|---|---|---|
| *Baseline UQ methods* | | | | | | |
| Prob | .442 ± .018 | .425 ± .031 | .162 ± .024 | .220 ± .014 | .254 ± .025 | .252 ± .077 |
| MTE | .534 ± .013 | .465 ± .029 | .161 ± .02 | .232 ± .034 | .253 ± .031 | .256 ± .052 |
| Perplexity | .422 ± .024 | .419 ± .022 | .157 ± .02 | .223 ± .023 | .252 ± .022 | .256 ± .062 |
| CCP | .533 ± .021 | **.478** ± .031 | .117 ± .022 | .303 ± .025 | .264 ± .034 | **.290** ± .062 |
| SAR | .533 ± .02 | .426 ± .028 | .176 ± .023 | .214 ± .018 | .033 ± .037 | .050 ± .055 |
| P(True) | -.076 ± .026 | -.155 ± .034 | -.161 ± .024 | -.090 ± .025 | -.046 ± .028 | -.047 ± .048 |
| SemanticEntropy | .449 ± .021 | .415 ± .022 | .166 ± .019 | .223 ± .022 | .254 ± .028 | .252 ± .051 |
| Lexical Similarity | .527 ± .017 | .427 ± .032 | .176 ± .025 | .127 ± .016 | .052 ± .032 | .172 ± .054 |
| EigValLaplacian | **.578** ± .014 | .472 ± .028 | .190 ± .036 | .134 ± .024 | .014 ± .027 | .010 ± .046 |
| NumSemSets | .556 ± .017 | .442 ± .028 | .123 ± .027 | .106 ± .019 | .046 ± .03 | .153 ± .042 |
| *Consistency-based UQ: multinomial vs. beamsearch versions* | | | | | | |
| Dissimilarity | .549 ± .019 | .415 ± .03 | .111 ± .022 | .068 ± .016 | .024 ± .04 | .070 ± .062 |
| Dissimilarity + beamsearch | .413 ± .023 | .321 ± .028 | .204 ↑± .016 | .273 ↑± .021 | .218 ↑± .028 | .085 ↑± .052 |
| Eccentricity | .540 ± .012 | .429 ± .029 | .167 ± .02 | .175 ± .026 | -.020 ± .034 | .094 ± .054 |
| Eccentricity + beamsearch | .441 ± .019 | .367 ± .035 | .235 ↑± .025 | **.314** ↑± .027 | .246 ↑± .024 | .108 ↑± .066 |
| EigVecDissimilarity | .561 ± .021 | .437 ± .027 | .169 ± .031 | .173 ± .02 | -.017 ± .032 | .095 ± .035 |
| EigVecDissimilarity + beamsearch | .478 ± .028 | .416 ± .032 | **.240** ↑± .021 | .308 ↑± .016 | .253 ↑± .028 | .113 ↑± .06 |
| CocoaMSP | .531 ± .023 | .456 ± .028 | .183 ± .021 | .198 ± .028 | .252 ± .03 | .266 ± .052 |
| CocoaMSP + beamsearch | .535 ↑± .02 | .473 ↑± .02 | .237 ↑± .02 | .287 ↑± .027 | **.282** ↑± .029 | .258 ± .072 |
| CocoaPPL | .523 ± .015 | .454 ± .025 | .174 ± .019 | .201 ± .023 | .247 ± .031 | .271 ± .039 |
| CocoaPPL + beamsearch | .522 ± .024 | .467 ↑± .024 | .222 ↑± .024 | .285 ↑± .019 | .277 ↑± .035 | .264 ± .048 |

Table 16: PRR (↑ is better) for 6 datasets with Llama 3.1 8B base. For each dataset, the top-1 method is **bold** and the second-best is underlined. Beam-guided and probability-weighted variants are marked with ↑ when they improve over their multinomial-sampling baseline.

| UQ Method | TriviaQA | Web Questions | CoQA | HotpotQA | Common senceQA | ARC-Challenge |
|---|---|---|---|---|---|---|
| *Baseline UQ methods* | | | | | | |
| Prob | .517 ± .019 | .414 ± .029 | .310 ± .022 | .213 ± .024 | .504 ± .029 | .505 ± .043 |
| MTE | .544 ± .018 | .420 ± .015 | .286 ± .022 | .327 ± .02 | .448 ± .029 | .511 ± .055 |
| Perplexity | .507 ± .015 | .441 ± .027 | .316 ± .031 | .375 ± .018 | .501 ± .027 | .570 ± .047 |
| CCP | .575 ± .016 | .420 ± .026 | .276 ± .024 | .247 ± .029 | .442 ± .023 | .446 ± .031 |
| SAR | .548 ± .017 | .452 ± .028 | .331 ± .03 | .263 ± .031 | .189 ± .021 | .330 ± .044 |
| P(True) | -.055 ± .021 | .059 ± .023 | -.020 ± .018 | -.223 ± .026 | .034 ± .024 | .292 ± .044 |
| SemanticEntropy | .538 ± .019 | .409 ± .023 | .330 ± .021 | .199 ± .024 | .492 ± .023 | .514 ± .05 |
| Lexical Similarity | .467 ± .018 | .396 ± .03 | .366 ± .024 | .289 ± .026 | .437 ± .028 | .511 ± .041 |
| EigValLaplacian | .569 ± .019 | .418 ± .022 | .377 ± .023 | .247 ± .025 | .449 ± .035 | .499 ± .047 |
| NumSemSets | .550 ± .014 | .409 ± .033 | .319 ± .019 | .241 ± .028 | .378 ± .025 | .477 ± .044 |
| *Consistency-based UQ: multinomial vs. beamsearch versions* | | | | | | |
| Dissimilarity | .576 ± .02 | .445 ± .024 | .473 ± .023 | .446 ± .02 | .449 ± .028 | .640 ± .056 |
| Dissimilarity + beamsearch | **.654** ↑± .017 | **.504** ↑± .023 | **.485** ↑± .019 | .424 ± .024 | .510 ↑± .023 | **.683** ↑± .044 |
| Eccentricity | .555 ± .016 | .404 ± .025 | .405 ± .023 | .297 ± .021 | .464 ± .028 | .591 ± .038 |
| Eccentricity + beamsearch | .613 ↑± .021 | .458 ↑± .019 | .429 ↑± .017 | .361 ↑± .023 | .512 ↑± .025 | .657 ↑± .031 |
| EigVecDissimilarity | .570 ± .015 | .452 ± .022 | .409 ± .019 | .289 ± .02 | .469 ± .04 | .587 ± .038 |
| EigVecDissimilarity + beamsearch | .630 ↑± .019 | .492 ↑± .022 | .427 ↑± .019 | .357 ↑± .02 | .506 ↑± .035 | .650 ↑± .032 |
| CocoaMSP | .595 ± .013 | .458 ± .021 | .463 ± .023 | .366 ± .021 | .510 ± .028 | .641 ± .038 |
| CocoaMSP + beamsearch | .631 ↑± .019 | .487 ↑± .023 | .465 ↑± .027 | .372 ↑± .027 | **.532** ↑± .022 | .639 ± .041 |
| CocoaPPL | .587 ± .017 | .464 ± .024 | .464 ± .023 | **.465** ± .02 | .501 ± .031 | .660 ± .034 |
| CocoaPPL + beamsearch | .616 ↑± .016 | .498 ↑± .029 | .459 ± .024 | .456 ± .018 | .525 ↑± .028 | .661 ↑± .046 |

Table 17: PRR (↑ is better) for 6 datasets with Llama 3.1 8B instruct. For each dataset, the top-1 method is **bold** and the second-best is underlined. Beam-guided and probability-weighted variants are marked with ↑ when they improve over their multinomial-sampling baseline.

| UQ Method | TriviaQA | Web Questions | CoQA | HotpotQA | Common senceQA | ARC-Challenge |
|---|---|---|---|---|---|---|
| *Baseline UQ methods* | | | | | | |
| Prob | .524 ± .023 | .357 ± .036 | .327 ± .021 | .213 ± .022 | .283 ± .026 | .363 ± .044 |
| MTE | .604 ± .015 | .424 ± .028 | .307 ± .02 | .253 ± .031 | .260 ± .027 | .339 ± .055 |
| Perplexity | .498 ± .018 | .367 ± .025 | .262 ± .025 | .221 ± .03 | .255 ± .028 | .332 ± .053 |
| CCP | .576 ± .023 | .406 ± .028 | .291 ± .018 | .265 ± .022 | .248 ± .034 | .402 ± .048 |
| SAR | .599 ± .021 | .420 ± .029 | .338 ± .024 | .236 ± .02 | .301 ± .025 | .418 ± .04 |
| P(True) | .236 ± .023 | .012 ± .031 | .018 ± .035 | .045 ± .024 | -.011 ± .024 | .135 ± .051 |
| SemanticEntropy | .591 ± .016 | .381 ± .027 | .335 ± .032 | .231 ± .029 | .301 ± .038 | .418 ± .061 |
| Lexical Similarity | .566 ± .023 | .395 ± .029 | .347 ± .024 | .232 ± .03 | .275 ± .032 | .380 ± .045 |
| EigValLaplacian | .615 ± .021 | .389 ± .026 | .355 ± .029 | .238 ± .023 | .252 ± .029 | .377 ± .051 |
| NumSemSets | .569 ± .021 | .363 ± .031 | .228 ± .03 | .180 ± .023 | .208 ± .035 | .368 ± .051 |
| *Consistency-based UQ: multinomial vs. beamsearch versions* | | | | | | |
| Dissimilarity | .616 ± .016 | .382 ± .031 | .349 ± .018 | .270 ± .021 | .277 ± .037 | .378 ± .061 |
| Dissimilarity + beamsearch | .662 ↑± .015 | .411 ↑± .029 | .358 ↑± .029 | **.349** ↑± .019 | .288 ↑± .032 | .434 ↑± .054 |
| Eccentricity | .598 ± .021 | .379 ± .032 | .319 ± .016 | .248 ± .031 | .273 ± .035 | .389 ± .058 |
| Eccentricity + beamsearch | .620 ↑± .016 | .396 ↑± .027 | .330 ↑± .021 | .281 ↑± .021 | .306 ↑± .031 | .451 ↑± .047 |
| EigVecDissimilarity | .611 ± .019 | .378 ± .033 | .325 ± .025 | .249 ± .029 | .264 ± .037 | .390 ± .061 |
| EigVecDissimilarity + beamsearch | .640 ↑± .017 | .425 ↑± .028 | .347 ↑± .027 | .291 ↑± .022 | **.318** ↑± .034 | **.461** ↑± .046 |
| CocoaMSP | .629 ± .018 | .409 ± .023 | .366 ± .03 | .278 ± .02 | .314 ± .029 | .426 ± .051 |
| CocoaMSP + beamsearch | **.665** ↑± .016 | **.428** ↑± .029 | **.378** ↑± .017 | .344 ↑± .019 | .302 ± .036 | .439 ↑± .041 |
| CocoaPPL | .626 ± .022 | .410 ± .03 | .354 ± .024 | .278 ± .024 | .299 ± .038 | .413 ± .056 |
| CocoaPPL + beamsearch | .653 ↑± .018 | .427 ↑± .032 | .356 ↑± .021 | .334 ↑± .018 | .285 ± .04 | .419 ↑± .056 |

Table 18: PRR (↑ is better) for 6 datasets with Qwen 3 8B base. For each dataset, the top-1 method is **bold** and the second-best is underlined. Beam-guided and probability-weighted variants are marked with ↑ when they improve over their multinomial-sampling baseline.

| UQ Method | TriviaQA | Web Questions | CoQA | HotpotQA | Common senceQA | ARC-Challenge |
|---|---|---|---|---|---|---|
| *Baseline UQ methods* | | | | | | |
| Prob | .617 ± .017 | .449 ± .025 | .267 ± .025 | .111 ± .033 | .337 ± .039 | .475 ± .085 |
| MTE | .602 ± .022 | .409 ± .027 | .267 ± .023 | .279 ± .023 | **.443** ± .044 | .444 ± .077 |
| Perplexity | .597 ± .018 | .426 ± .028 | .278 ± .023 | .256 ± .026 | .294 ± .045 | .381 ± .065 |
| CCP | .640 ± .018 | .406 ± .028 | .213 ± .028 | .153 ± .025 | .296 ± .048 | .421 ± .09 |
| SAR | .617 ± .023 | .457 ± .023 | .323 ± .03 | .243 ± .028 | .220 ± .042 | .317 ± .066 |
| P(True) | .322 ± .021 | .282 ± .025 | .005 ± .031 | .168 ± .024 | -.043 ± .045 | -.074 ± .069 |
| Semantic Entropy | .549 ± .018 | .411 ± .02 | .247 ± .025 | .173 ± .023 | .230 ± .026 | .305 ± .058 |
| Lexical Similarity | .595 ± .023 | .430 ± .024 | .338 ± .019 | .310 ± .025 | .367 ± .042 | .508 ± .076 |
| EigValLaplacian | .602 ± .015 | .423 ± .027 | .301 ± .027 | .284 ± .028 | .349 ± .032 | .475 ± .081 |
| NumSemSets | .593 ± .016 | .403 ± .029 | .268 ± .024 | .250 ± .023 | .311 ± .039 | .367 ± .069 |
| *Consistency-based UQ: multinomial vs. beamsearch versions* | | | | | | |
| Dissimilarity | .668 ± .014 | .462 ± .024 | .406 ± .023 | **.531** ± .017 | .315 ± .038 | .476 ± .086 |
| Dissimilarity + beamsearch | **.680** ↑± .019 | .484 ↑± .024 | **.409** ↑± .03 | .504 ± .019 | .335 ↑± .044 | .457 ± .088 |
| Eccentricity | .615 ± .016 | .416 ± .023 | .320 ± .022 | .319 ± .024 | .266 ± .053 | .440 ± .068 |
| Eccentricity + beamsearch | .640 ↑± .013 | .437 ↑± .025 | .368 ↑± .02 | .407 ↑± .026 | .243 ± .04 | .366 ± .072 |
| EigVecDissimilarity | .628 ± .014 | .454 ± .028 | .325 ± .026 | .314 ± .027 | .373 ± .035 | .456 ± .071 |
| EigVecDissimilarity + beamsearch | .660 ↑± .016 | .460 ↑± .024 | .380 ↑± .025 | .394 ↑± .025 | .353 ± .045 | .453 ± .086 |
| CocoaMSP | .667 ± .019 | .492 ± .019 | .385 ± .025 | .320 ± .025 | .378 ± .028 | **.523** ± .071 |
| CocoaMSP + beamsearch | .678 ↑± .018 | **.498** ↑± .028 | .391 ↑± .02 | .378 ↑± .029 | .385 ↑± .037 | .510 ± .079 |
| CocoaPPL | .665 ± .015 | .478 ± .019 | .388 ± .035 | .397 ± .024 | .353 ± .038 | .484 ± .081 |
| CocoaPPL + beamsearch | .667 ↑± .016 | .486 ↑± .021 | .387 ± .036 | .437 ↑± .026 | .339 ± .044 | .450 ± .06 |

Table 19: PRR (↑ is better) for 6 datasets with Qwen 3 8B instruct. For each dataset, the top-1 method is **bold** and the second-best is underlined. Beam-guided and probability-weighted variants are marked with ↑ when they improve over their multinomial-sampling baseline.

| UQ Method | TriviaQA | Web Questions | CoQA | HotpotQA | Common senceQA | ARC-Challenge |
|---|---|---|---|---|---|---|
| *Baseline UQ methods* | | | | | | |
| Prob | .564 ± .017 | .353 ± .032 | .215 ± .02 | .250 ± .026 | .174 ± .034 | .181 ± .078 |
| MTE | .564 ± .018 | .345 ± .028 | .164 ± .025 | .251 ± .028 | .183 ± .03 | .272 ± .095 |
| Perplexity | .491 ± .023 | .341 ± .036 | .169 ± .026 | .250 ± .028 | .175 ± .037 | .229 ± .058 |
| CCP | .563 ± .02 | .383 ± .029 | .169 ± .018 | .258 ± .029 | .173 ± .034 | .202 ± .068 |
| SAR | .590 ± .016 | .425 ± .036 | .146 ± .026 | .159 ± .029 | .201 ± .033 | .233 ± .051 |
| P(True) | -.105 ± .023 | -.222 ± .035 | -.126 ± .017 | .018 ± .021 | -.083 ± .03 | -.164 ± .071 |
| Semantic Entropy | .597 ± .016 | .404 ± .034 | .214 ± .022 | .231 ± .026 | .174 ± .041 | .176 ± .08 |
| Lexical Similarity | .530 ± .023 | .425 ± .029 | .193 ± .031 | .101 ± .025 | .121 ± .039 | .053 ± .06 |
| EigValLaplacian | .626 ± .015 | .417 ± .04 | .196 ± .026 | .083 ± .024 | .134 ± .031 | .134 ± .066 |
| NumSemSets | .608 ± .021 | .437 ± .036 | .110 ± .019 | .096 ± .024 | .113 ± .041 | .154 ± .065 |
| *Consistency-based UQ: multinomial vs. beamsearch versions* | | | | | | |
| Dissimilarity | .588 ± .017 | .382 ± .03 | .165 ± .02 | .187 ± .025 | **.246** ± .038 | **.394** ± .072 |
| Dissimilarity + beamsearch | .637 ↑± .018 | .386 ↑± .026 | .269 ↑± .019 | .264 ↑± .026 | .213 ± .031 | .362 ± .083 |
| Eccentricity | .565 ± .019 | .367 ± .034 | .167 ± .025 | .125 ± .023 | .150 ± .026 | .132 ± .078 |
| Eccentricity + beamsearch | .600 ↑± .016 | .392 ↑± .034 | .288 ↑± .029 | .291 ↑± .022 | .211 ↑± .035 | .285 ↑± .084 |
| EigVecDissimilarity | .590 ± .024 | .385 ± .031 | .169 ± .026 | .121 ± .032 | .143 ± .033 | .131 ± .066 |
| EigVecDissimilarity + beamsearch | **.645** ↑± .016 | **.439** ↑± .032 | **.328** ↑± .019 | **.297** ↑± .017 | .242 ↑± .029 | .306 ↑± .058 |
| CocoaMSP | .607 ± .015 | .394 ± .03 | .204 ± .016 | .272 ± .023 | .230 ± .042 | .298 ± .061 |
| CocoaMSP + beamsearch | .635 ↑± .02 | .404 ↑± .024 | .263 ↑± .023 | .282 ↑± .025 | .206 ± .029 | .290 ± .061 |
| CocoaPPL | .581 ± .02 | .389 ± .032 | .179 ± .024 | .272 ± .022 | .232 ± .032 | .309 ± .082 |
| CocoaPPL + beamsearch | .609 ↑± .02 | .395 ↑± .031 | .233 ↑± .025 | .282 ↑± .026 | .207 ± .03 | .299 ± .084 |

# E    DETAILED DESCRIPTION OF UNCERTAINTY QUANTIFICATION METHODS

In this section, we describe the uncertainty quantification methods used in our experiments.

**Sequence Probability (Prob)** is the most straightforward approach to uncertainty quantification. We define it formally as the negative log-probability of the generating sequence:

$$U_{\text{SP}}(\mathbf{y} \mid \mathbf{x}) = -\log P(\mathbf{y} \mid \mathbf{x}). \tag{27}$$

**Mean Token Entropy (MTE)** measures an average entropy of tokens in a sequence:

$$U_{\text{MTE}}(\mathbf{y} \mid \mathbf{x}) = \frac{1}{L} \sum_{l=1}^{L} \mathcal{H}(y_l \mid \mathbf{y}_{<l}, \mathbf{x}), \tag{28}$$

where $\mathcal{H}(y_l \mid \mathbf{y}_{<l}, \mathbf{x}) = -\sum_v P(y_l = v \mid \mathbf{y}_{<l}, \mathbf{x}) \log P(y_l = v \mid \mathbf{y}_{<l}, \mathbf{x})$.

**Perplexity** computes negative average log-likelihood of tokens in a sequence:

$$U_{\text{PPL}}(\mathbf{y} \mid \mathbf{x}) = -\frac{1}{L} \log P(\mathbf{y} \mid \mathbf{x}), \tag{29}$$

**Claim Conditioned Probability (CCP)**, introduced in (Fadeeva et al., 2024), measures uncertainty on a claim level by perturbing claim's tokens with alternative generations:

$$U_{\text{CCP}}(C \mid \mathbf{x}) = 1 - \prod_{j \in C} \text{CCP}(y_j \mid y_{<j}, \mathbf{x}). \tag{30}$$

Where $\text{CCP}(y_j \mid \mathbf{y}_{<j}, \mathbf{x}) = \frac{\sum_{k:\text{NLI}(y_j^k, y_j) = 'e'} P(y_j^k \mid \mathbf{y}_{<j}, \mathbf{x})}{\sum_{k:\text{NLI}(y_j^k, y_j) \in \{'e', 'c'\}} P(y_j^k \mid \mathbf{y}_{<j}, \mathbf{x})}$

**Shifting Attention to Relevance (SAR)** is a method combining TokenSAR and SentenceSAR, as introduced by Duan et al. (2024). SentenceSAR is defined as follows:

$$U_{\text{SentSAR}}(\mathbf{x}) = -\frac{1}{M} \sum_{i=1}^{M} \log\Big( p(\mathbf{y}^{(i)} \mid \mathbf{x}) + \frac{1}{t} \text{R}_S(\mathbf{y}^{(i)}, \mathbf{x}) \Big), \tag{31}$$

Here, $\text{R}_S(\mathbf{y}^{(j)}, \mathbf{x}) = \sum_{k \neq j} s(\mathbf{y}^{(j)}, \mathbf{y}^{(k)}) p(\mathbf{y}^{(k)} \mid \mathbf{x})$. To obtain SAR score, the generative probability $p(\mathbf{y} \mid \mathbf{x})$ is replaced with relevance-reweighted probability on a sequence level. *TokenSAR* is defined as:

$$U_{\text{TokenSAR}}(\mathbf{x}) = -\sum_{l=1}^{L} \tilde{R}_T(y_l, \mathbf{y}, \mathbf{x}) \log P(y_l \mid \mathbf{y}_{<l}, \mathbf{x}), \tag{32}$$

where $R_T(\cdot)$ denotes some token relevance function and relevance weight for token $y_l$ is given by $\tilde{R}_T(y_k, \mathbf{y}, \mathbf{x}) = \frac{R_T(y_k, \mathbf{y}, \mathbf{x})}{\sum_{l=1}^{L} R_T(y_l, \mathbf{y}, \mathbf{x})}$ .

**P(True)**, introduced in (Kadavath et al., 2022), evaluates the confidence in a generation by asking the model the original question and answer, then asking if it is true or false. We then use the negative log-probability of the token "True" as an uncertainty score.

**Lexical Similarity**, introduced in (Fomicheva et al., 2020), measures average pairwise similarity between $M$ sampled generations using some similarity function $s(\mathbf{y}, \mathbf{y}')$:

$$U_{\text{LSRL}}(\mathbf{x}) = 1 - \frac{2}{M(M-1)} \sum_{i<j} s(\mathbf{y}^{(i)}, \mathbf{y}^{(j)}). \tag{33}$$

**Number of Semantic Sets**, introduced in (Lin et al., 2024), estimates how many distinct meanings the model produces by clustering its outputs with an NLI model. Two answers are placed in the same cluster if they mutually entail each other more than they contradict and the final number of distinct clusters serves as an uncertainty score $U_{\text{NumSemSets}}$.

**Sum of Eigenvalues of Laplacian**, introduced in (Lin et al., 2024), constructs a similarity matrix among the sampled outputs and computes a uncertainty score from the eigenvalues of the Laplacian of that similarity matrix:

$$U_{\text{EigV}}(\mathbf{x}) = \sum_{i=1}^{M} \max\big(0, 1 - \lambda_i(\mathbf{x})\big). \tag{34}$$

## F    COMPUTATIONAL BUDGET

All experiments were run on $2\times$NVIDIA A100 (80 GB). Evaluating a single model across all six datasets took approximately 2 wall-clock days on this setup (4 GPU-days); with six models, this amounts to 12 wall-clock days (24 GPU-days). Additional ablations (sampling strategies, top-1 beam scoring, and other objectives) required a further 5 wall-clock days on the same hardware (10 GPU-days). In total, the study used about 34 GPU-days.

## G    THE USAGE OF LLMS

In this study, large language models are examined primarily as the focus of analysis. For practical tasks such as programming and writing, we also make limited use of LLM-based assistants (e.g., ChatGPT) to support grammar correction and code debugging, with all usage carefully monitored by humans.

