# OpenReview forum: "Don't Throw Away Your Beams: Improving Consistency-based Uncertainties in LLMs via Beam Search"
_ICLR.cc/2026/Conference — ICLR 2026 Poster_

### Official Review · Reviewer_fdB8 · 2025-10-27

**Soundness:** 4
**Presentation:** 3
**Contribution:** 3
**Rating:** 8
**Confidence:** 5

**Summary:**

The authors propose replacing multinomial sampling with beam search sampling for consistency-based UQ methods. They show that it's even theoretically less erroneous in low-sampling regimes. They verify the proposed method's effectiveness on various datasets and various methods.

**Strengths:**

- The paper focuses on an important topic with a neat, focused contribution.
- The paper is well written.
- The proposed replacement of multinomial sampling with beam search definitely makes sense.
- The idea is supported by a clean, small, but necessary theory.
- Very good ablations and additional experiments with different temperatures and sampling ideas.
- Clear demonstration of the effectiveness of the idea on various consistency-based methods.

**Weaknesses:**

- I don't see any major weaknesses, but the impact of the work is probably limited to the UQ community.

**Questions:**

- Do you have any results for when you combine your idea with other methods that do sampling but use probabilities, such as SE and SAR?

---

> ### Author Response · Authors · 2025-11-20
>
> We appreciate the reviewer’s recognition of our work and are glad to address the remaining questions.
>
> **Q: Do you have any results for when you combine your idea with other methods that do sampling but use probabilities, such as SE and SAR?**
>
> We already include beam-search variants of Semantic Entropy and the Degree Matrix methods in Appendix A3 (Table 6), where beam search provides clear improvements for Semantic Entropy, while the Degree Matrix methods show little to no change. We additionally ran a beam-search variant of SAR, and it also follows the same pattern (see attached table). We will incorporate these results into the final version.
>
> | UQ Method                       | TriviaQA | Web Questions | CoQA | HotpotQA | CommonsenseQA | ARC-Challenge | Mean |
> |---------------------------------|----------|----------------|------|----------|----------------|----------------|-------|
> | Semantic Entropy                | .622     | .505          | .301 | .140     | .407          | .431          | .401 |
> | Semantic Entropy + beamsearch   | .685$\uparrow$ | .614$\uparrow$ | .365$\uparrow$ | .278$\uparrow$ | .436$\uparrow$ | .454$\uparrow$ | .472$\uparrow$ |
> | Degree Matrix                   | .682     | .605          | .385 | .311     | .409          | .419          | .469 |
> | Degree Matrix + beamsearch      | .673     | .642$\uparrow$ | .328 | .244     | .444$\uparrow$ | .473$\uparrow$ | .467 |
> | SAR                             | .656     | .571          | .347 | .296     | .183          | .264          | .386 |
> | SAR + beamsearch                | .671$\uparrow$ | .589$\uparrow$ | .329 | .266     | .209$\uparrow$ | .269$\uparrow$ | .372 |
> | Dissimilarity                   | .755 | .715      | .578 | .626 | .561          | .545          | .630 |
> | Dissimilarity + beamsearch      | **.766$\uparrow$** | **.722$\uparrow$** | **.600$\uparrow$** | .611 | **.595$\uparrow$** | .604$\uparrow$ | **.650$\uparrow$** |
> | Eccentricity                    | .714     | .653          | .459 | .453     | .549          | .549          | .563 |
> | Eccentricity + beamsearch       | .739$\uparrow$ | .633 | .505$\uparrow$ | .514 | .590$\uparrow$ | **.636$\uparrow$** | .603$\uparrow$ |
> | EigVecDissimilarity             | .738     | .661          | .443 | .448     | .512          | .562          | .561 |
> | EigVecDissimilarity + beamsearch| .753$\uparrow$ | .668$\uparrow$ | .497$\uparrow$ | .487 | .562$\uparrow$ | .621$\uparrow$ | .598$\uparrow$ |
> | CocoaMSP                        | .738     | .666          | .509 | .430     | .583          | .595          | .587 |
> | CocoaMSP + beamsearch           | .747$\uparrow$ | .679$\uparrow$ | .548$\uparrow$ | .523 | .586$\uparrow$ | .606$\uparrow$ | .615$\uparrow$ |
> | CocoaPPL                        | .739     | .678          | .548 | .625     | .580          | .595          | .628 |
> | CocoaPPL + beamsearch           | .748$\uparrow$ | .694$\uparrow$ | .577$\uparrow$ | **.681$\uparrow$** | .582$\uparrow$ | .610$\uparrow$ | .649$\uparrow$ |

---

> ### Author Response · Authors · 2025-11-25
>
> Dear reviewer,
> Thank you again for your time and thoughtful feedback. With one week remaining in the discussion period, we kindly ask you to review our responses. Our rebuttal addresses the concerns you raised in your review. If there are no concerns remaining, we would greatly appreciate it if you could reconsider your scores accordingly. Should any questions remain, we would be happy to provide further clarification.

---

> > ### Comment · Reviewer_fdB8 · 2025-11-26
> >
> > Thanks to the authors for the additional results. I maintain my positive score.

---

### Official Review · Reviewer_1K6A · 2025-10-30

**Soundness:** 3
**Presentation:** 3
**Contribution:** 3
**Rating:** 8
**Confidence:** 3

**Summary:**

The paper targets uncertainty quantification (UQ) for short-form QA where consistency-based methods aggregate multiple generations to estimate confidence. It argues that the standard practice to generate samples via multinomial sampling under a small budget produces many duplicates in peaky distributions. This results in high variance and unstable UQ. The authors propose to use beam-search with probability weighting to obtain samples for the consistency estimator. Experiments across several QA datasets report consistent gains in UQ metrics.

**Strengths:**

The paper clearly motivates the problem and proposes a simple drop-in replacement for multinomial sampling in consistency-based UQ via probability-weighted beam search. It provides a bias–variance analysis with an interpretable beam-mass condition. Experiments on multiple short-form QA datasets show consistent gains, including extensive ablation studies. Moreover, the probability-weighted beams consistently improve other UQ baselines (e.g., semantic entropy), suggesting broader applicability.

**Weaknesses:**

The improvements appear strongest for short answers, the coverage of probability mass by top-M beams degrades with length (Fig. 3). It’s unclear whether the advantages vanish or even reverse for longer outputs. An additional discussion or experiments on benchmarks with longer outputs could help to clarify.

**Questions:**

* For what answer lengths and budget M does beam-weighted consistency cease to outperform sampling? Any failure cases?
* Could you discuss whether there are use cases in which beam search produces suboptimal predictions which may lead to unreliable uncertainty estimates?
* Figure 4 indicates that even for M=1, the PRR of the beam search is comparable with the multinomial sampling with up to M=15 (dissimilarity) and M=5 (eccentricity). Could you explain why the beam search with such small beam width is performing so well?
* The related works section is relatively short. Work of Hashimoto et al. (2025) is mentioned where different decoding strategies are compared in terms of UQ for different tasks. How is the beam search approach performing in those evaluations? Is the main difference and gain in this paper the combination of beam search with weighted consistency scores?

---

> ### Author Response · Authors · 2025-11-20
>
> **Q1: The improvements appear strongest for short answers, the coverage of probability mass by top-M beams degrades with length (Fig. 3). It’s unclear whether the advantages vanish or even reverse for longer outputs. An additional discussion or experiments on benchmarks with longer outputs could help to clarify.**
>
> The proposed UQ method can in principle be used for both short- and long-form generations. However, our theoretical results and experimental findings indicate that it performs better on short answer QA. In the case of longer generations with autoregressive probabilistic models, less probability mass will be observed within a limited number of sequences generated by beam search. This in turn reduces the likelihood that probability mass covered by beam search will exceed the threshold defined by Eq. 7. Empirical study of the prevalence of successful coverage is reported on Fig. 3, supporting this analysis. This explains why the performance gap between beam search and sampling reduces as generations get longer, as indicated by Fig. 5.
>
> To investigate performance of our approach on longer generations we also conduct experiments on the GSM8k dataset. We use reasoning elicitation with prompt, compute per-step uncertainty and aggregate it over steps via mean-pooling.
>
> | Method                                    | Value              |
> |-------------------------------------------|--------------------|
> | MSP                                       | .347               |
> | MTE                                       | .336               |
> | Perplexity                                | .332               |
> | CCP                                       | .303               |
> | P(True)                                   | .271               |
> | SemanticEntropy                           | .340               |
> | Lexical Similarity                        | .376               |
> | EigValLaplacian                           | .350               |
> | NumSemSets                                | .348               |
> | Dissimilarity                             | .354               |
> | Dissimilarity + beamsearch                | **.373**$\uparrow$ |
> | Eccentricity                              | .342               |
> | Eccentricity + beamsearch                 | .346$\uparrow$     |
> | EigVecDissimilarity                       | .337               |
> | EigVecDissimilarity + beamsearch          | .347$\uparrow$     |
> | CocoaMSP                                  | .360               |
> | CocoaMSP + beamsearch                     | .369$\uparrow$     |
> | CocoaPPL                                  | .337               |
> | CocoaPPL + beamsearch                     | .358$\uparrow$     |
>
>
> This supports our claim that beam search is preferable to multinomial sampling from the UQ standpoint, as it facilitates better UQ performance on short form tasks and works either better or identical to sampling on a long-form task.
>
> **Q2: For what answer lengths and budget M does beam-weighted consistency cease to outperform sampling? Any failure cases? Could you discuss whether there are use cases in which beam search produces suboptimal predictions which may lead to unreliable uncertainty estimates?**
>
> On all our short-length datasets beam-weighted consistency outperforms multinomial sampling. The average response length for these datasets is 5.2 tokens.
> On long-form generations, beam search produces more similar samples, while multinomial sampling produces more diverse candidates. See the example below:
>
> ---
>
> ### Question:
> *What are the last six words of Rudyard Kipling's *If*?*
>
> ### Gold Answer:
> *you'll be a man, my son*
>
> ### Beam Search Candidates:
> 1. if you can fill the unforgiving minute *(p=0.030)*
> 2. if you can talk with crowds and keep your virtue *(p=0.015)*
> 3. if you can fill the unforgiving minute with *(p=0.012)*
> 4. then you are the man, my friend *(p=0.005)*
> 5. If you can fill the unforgiving minute *(p=0.005)*
> 6. if you can fill the unforgiving minute, *(p=0.003)*
> 7. …
>
> ### Sampling Candidates:
> 1. And most of all -if you can bear to keep still *(p=2.275e-09)*
> 2. and all shall know me by the way *(p=1.205e-09)*
> 3. as yet untried. *(p=2.974e-11)*
> 4. you remain unimpeachable *(p=5.354-16)*
> 5. and stoop to compromise *(p=1.412e-08)*
> 6. and yet supposing you win *(p=1.011e-08)*
> 7. …
>
> ### Dissimilarity Scores:
> - Dissimilarity (Beam Search): *0.604*
> - Dissimilarity (Sampling): *0.860*
>
> ---
>
> In this example, we see that the beam search produces more similar responses than the multinomial sampling, and the resulting dissimilarity is higher for the sampling option ($0.604$ vs $0.860$). We expect high uncertainty because all the answers are incorrect. The total beam mass of $0.078$ is significantly less than the theoretical limit of $0.842$ (when, in theory, beam search is better than sampling).
>
> Furthermore, in Figure 3 of the paper, we showed that the more tokens there are in the generation, the more difficult it is to accumulate beam search mass.

---

> ### Author Response · Authors · 2025-11-20
>
> **Q3: Figure 4 indicates that even for M=1, the PRR of the beam search is comparable with the multinomial sampling with up to M=15 (dissimilarity) and M=5 (eccentricity). Could you explain why the beam search with such a small beam width is performing so well?**
>
> For the original Figure 4, we first generated 15 samples/beams and then reused the first $M = 1, \dots, 15$ to compute the UQ scores. This inadvertently inflated the performance at very small $M$, since e.g. “M=1” for beam search was effectively the top-1 result from a 15-beam run rather than a true 1-beam decoding. This artifact only affects Fig. 4, all other results remain unchanged. We thank the reviewer for pointing this out.
>
> We have re-computed the figure using exactly $M$ beams for each $M \in$ {$1, \dots, 15$} and updated both Fig. 4 and corresponding results discussion (Section 4.3.1) in the new paper PDF. The updated plot shows the expected behavior: beam search still consistently outperforms sampling across all $M>1$, but for $M=1$ the beam search performance degrades to approximately zero PRR.
>
> **Q4: The related works section is relatively short. Work of Hashimoto et al. (2025) is mentioned where different decoding strategies are compared in terms of UQ for different tasks. How is the beam search approach performing in those evaluations? Is the main difference and gain in this paper the combination of beam search with weighted consistency scores?**
>
> We agree that the related works section is short. The main reason for it is that a lot of background and related works were mentioned in Section 2 (Background and Motivation) and we had some space limitations. When it comes to the work of Hashimoto et al. (2025), the main goal of this paper was to examine the impact of the decoding strategies on produced sequence probability and mean token entropy scores. The authors found that sequence probabilities/ token entropy produced under Beam Search can sometimes underperform relative to greedy. We would like to emphasise that while relevant, Hashimoto et al. (2025) explores only 2 token-probabilities based methods which require single generation, while our focus on the consistency-based uncertainty estimation.
>
> We expanded the discussion of this work in the updated PDF version, as well as the section about the uncertainty-aware decoding strategies.

---

> ### Author Response · Authors · 2025-11-25
>
> Dear reviewer,
> Thank you again for your time and thoughtful feedback. With one week remaining in the discussion period, we kindly ask you to review our responses. Our rebuttal addresses the concerns you raised in your review. If there are no concerns remaining, we would greatly appreciate it if you could reconsider your scores accordingly. Should any questions remain, we would be happy to provide further clarification.

---

> > ### Comment · Reviewer_1K6A · 2025-11-26
> >
> > Thank you for your rebuttal. All my questions are clarified and I have no further issues. I am keeping my positive score.

---

> > > ### Author Response · Authors · 2025-11-26
> > > **Thanks!**
> > >
> > > Thanks for your positive assessment! Given that all your questions have been clarified,  can you possibly increase the confidence of your assessment?

---

### Official Review · Reviewer_jkE6 · 2025-10-31

**Soundness:** 3
**Presentation:** 3
**Contribution:** 2
**Rating:** 4
**Confidence:** 4

**Summary:**

The paper presents a new uncertainty quantification scheme for LLMs that leverages the sequence probabilities of the multiple beams obtained during the generation phase of the response. This is in contrast to existing consistency-based approaches that typically rely on generating multiple sample responses from the distribution represented by the LLM. The basic idea behind the proposed uncertainty estimator is to weight the disimilarity measure of each sequence by it' s corresponding normalised probability. Furthermore, the paper also shows that several consistency based estimators such as those based on eccentricity or eigenvectors dissimilarity can be expressed in terms of the probabilities of the beams. The experimental evaluation is restricted to QA benchmark datasets and 3 open models from the Llama, Qwen and Gemma families. The results show indeed improved performance over the corresponding UQ schemes that rely on multinomial sampling.

**Strengths:**

- Uncertainty quantification for LLM is currently a hot topic and therefore advances in this area are warranted.

- The quality of the presentation is overall quite good and therefore the paper is relatively easy to follow even by readers outside this research area. Most of the technical details presented in the paper are discussed in a relatively clear manner.

**Weaknesses:**

- The experimental evaluation considers only one generative task, namely QA, where responses are typically fairly short. LLMs are applied to a much wider range of tasks, and therefore it would be very interesting to see how the proposed approach performs in other tasks, such as for example summarisation or text-to-SQL translation.

- The performance improvements appear to be just marginal over the standard multinomial sampling e.g., 0.543 vs 0.505 in Table 3. Standard deviations appear to be missing from the table, and these need to be included to account for the noise. Also, for such mediocre performance bumps, statistical significance tests are mandatory.

- The proposed method assumes access to sequence probabilities and therefore it is not applicable to closed models such as GPT or Gemini. I think this limitation should be made clearer in the presentation.

**Questions:**

- I would be interested to get your perspective if I am to apply your method in summarisation tasks where sequences are longer. How would that affect the sequence probabilities?

---

> ### Author Response · Authors · 2025-11-20
>
> We are grateful to the reviewer for their careful evaluation and helpful insights. Below, we respond to each comment and clarify the points raised.
>
> **Q1: The experimental evaluation considers only one generative task, namely QA, where responses are typically fairly short. LLMs are applied to a much wider range of tasks, and therefore it would be very interesting to see how the proposed approach performs in other tasks, such as for example summarisation or text-to-SQL translation.**
>
> The proposed UQ method can in principle be used for both short- and long-form generations. However, our theoretical results and experimental findings indicate that it performs better on short answer QA. In the case of longer generations with autoregressive probabilistic models, less probability mass will be observed within a limited number of sequences generated by beam search. This in turn reduces the likelihood that total probability mass covered by beam search will exceed the threshold defined by Eq. 7. Empirical study of the prevalence of successful coverage is reported on Fig. 3, supporting this analysis. This explains why the performance gap between beam search and sampling approaches reduces as generations get longer, as indicated by Fig. 5.
>
> To further investigate performance of our approach on longer generations we also conduct experiments on the GSM8k dataset. We use reasoning elicitation with prompt, compute per-step uncertainty and aggregate it over steps via mean-pooling.
>
> | Method                                    | Value              |
> |-------------------------------------------|--------------------|
> | MSP                                       | .347               |
> | MTE                                       | .336               |
> | Perplexity                                | .332               |
> | CCP                                       | .303               |
> | P(True)                                   | .271               |
> | SemanticEntropy                           | .340               |
> | Lexical Similarity                        | .376               |
> | EigValLaplacian                           | .350               |
> | NumSemSets                                | .348               |
> | Dissimilarity                             | .354               |
> | Dissimilarity + beamsearch                | **.373**$\uparrow$ |
> | Eccentricity                              | .342               |
> | Eccentricity + beamsearch                 | .346$\uparrow$     |
> | EigVecDissimilarity                       | .337               |
> | EigVecDissimilarity + beamsearch          | .347$\uparrow$     |
> | CocoaMSP                                  | .360               |
> | CocoaMSP + beamsearch                     | .369$\uparrow$     |
> | CocoaPPL                                  | .337               |
> | CocoaPPL + beamsearch                     | .358$\uparrow$     |
>
>
> This supports our claim that beam search is generally preferable to multinomial sampling from the UQ standpoint, as it facilitates better UQ performance on short form tasks and works either better or identical to sampling on a long-form generation task.
> Regarding summarisation specifically, we have found it to be a tricky subject for UQ evaluation, as estimating the quality of the given summary in a principled way is a hard task in and of itself. As most UQ performance metrics rely on a meaningful response quality indicator, it becomes unclear how to provide informative comparison of UQ methods in this setting.

---

> > ### Author Response · Authors · 2025-11-20
> >
> > **Q2: The performance improvements appear to be just marginal .. standard deviations appear to be missing from the table.**
> >
> > Thank you for the comment. We have computed standard deviations for Table 3 (see table below), which is approximately $0.02$ for nearly all entries. The improvements (i.e. $0.505 \rightarrow 0.543$ with STDs $0.018$ and $0.019$), are larger than the corresponding STDs, so the gains are likely are not due to noise. We chose large datasets (1,000-2,000 examples) to keep variance low.
> >
> > | Method                          | Llama 3.1 8B base       | Llama 3.1 8B instruct       | Gemma 3 4B base        | Gemma 3 4B instruct     | Qwen 3 8B base         | Qwen 3 8B instruct     |
> > |---------------------------------|--------------------------|------------------------------|--------------------------|--------------------------|--------------------------|--------------------------|
> > | MSP                             | .410 ± .019              | .344 ± .031                  | .471 ± .023              | .292 ± .022              | .376 ± .03               | .289 ± .067              |
> > | MTE                             | .422 ± .016              | .364 ± .026                  | .476 ± .022              | .317 ± .028              | .407 ± .032              | .297 ± .064              |
> > | Perplexity                      | .452 ± .02               | .323 ± .027                  | .525 ± .024              | .288 ± .025              | .372 ± .033              | .276 ± .052              |
> > | CCP                             | .401 ± .02               | .364 ± .029                  | .492 ± .022              | .331 ± .026              | .355 ± .034              | .291 ± .06               |
> > | SAR                             | .352 ± .02               | .385 ± .029                  | .386 ± .026              | .239 ± .024              | .363 ± .033              | .292 ± .052              |
> > | P(True)                         | .015 ± .023              | .072 ± .03                   | .093 ± .026              | -.096 ± .024             | .110 ± .037              | -.114 ± .055             |
> > | SemanticEntropy                 | .414 ± .019              | .376 ± .025                  | .401 ± .023              | .293 ± .024              | .319 ± .031              | .299 ± .058              |
> > | Lexical Similarity              | .411 ± .02               | .366 ± .025                  | .426 ± .023              | .247 ± .023              | .425 ± .034              | .237 ± .055              |
> > | EigValLaplacian                 | .426 ± .016              | .371 ± .028                  | .437 ± .023              | .233 ± .025              | .406 ± .032              | .265 ± .056              |
> > | NumSemSets                      | .396 ± .018              | .319 ± .031                  | .418 ± .024              | .238 ± .023              | .365 ± .033              | .253 ± .052              |
> > |   Dissimilarity                 | .505 ± .018              | .379 ± .028                  | .630 ± .021              | .206 ± .019              | .477 ± .037              | .327 ± .066              |
> > | Dissimilarity + beamsearch      | **.543 ± .019** $\uparrow$ | .417 ± .026 $\uparrow$        | **.650 ± .022** $\uparrow$ | .252 ± .022 $\uparrow$    | **.478 ± .031**          | .355 ± .062 $\uparrow$    |
> > | Eccentricity                    | .453 ± .016              | .368 ± .029                  | .563 ± .021              | .231 ± .025              | .396 ± .035              | .251 ± .058              |
> > | Eccentricity + beamsearch       | .505 ± .017 $\uparrow$     | .397 ± .029 $\uparrow$        | .603 ± .023 $\uparrow$     | .285 ± .024 $\uparrow$     | .410 ± .03 $\uparrow$     | .345 ± .061 $\uparrow$     |
> > | EigVecDissimilarity             | .463 ± .019              | .370 ± .028                  | .561 ± .026              | .236 ± .025              | .425 ± .032              | .256 ± .051              |
> > | EigVecDissimilarity + beamsearch| .510 ± .021 $\uparrow$     | .414 ± .028 $\uparrow$        | .598 ± .022 $\uparrow$     | .301 ± .019 $\uparrow$     | .450 ± .033 $\uparrow$     | **.376 ± .057** $\uparrow$ |
> > | CocoaMSP                        | .505 ± .018              | .404 ± .025                  | .587 ± .023              | .314 ± .024              | .461 ± .031              | .334 ± .054              |
> > | CocoaMSP + beamsearch           | .521 ± .019 $\uparrow$     | **.426 ± .024** $\uparrow$     | .615 ± .021 $\uparrow$     | **.345 ± .026** $\uparrow$ | .473 ± .035 $\uparrow$     | .347 ± .061 $\uparrow$     |
> > | CocoaPPL                        | .523 ± .017              | .397 ± .026                  | .628 ± .024              | .312 ± .023              | .461 ± .034              | .327 ± .055              |
> > | CocoaPPL + beamsearch           | .536 ± .02 $\uparrow$      | .412 ± .027 $\uparrow$        | .649 ± .023 $\uparrow$     | .339 ± .021 $\uparrow$     | .461 ± .035 $\uparrow$     | .337 ± .057 $\uparrow$     |

---

> ### Author Response · Authors · 2025-11-20
>
> **Q3: The proposed method assumes access to sequence probabilities and therefore it is not applicable to closed models such as GPT or Gemini. I think this limitation should be made clearer in the presentation.**
>
>
> It is true that in our current experiments we assume access to the model probabilities. We will include this into the limitations section for the final version. However, we would like to note that the methods could be extended into a black-box settings using empirical probability estimates. In addition, white-box models are widely used, so developing methods tailored to such settings is necessary. Several of the recent SOTA methods for UQ in LLMs also require access to token probabilities - such as Semantic Entropy [1], Shifting Attention to Relevance [2] or Claim Conditioned Probability [3]. Semantic Entropy in particular was implemented using empirical probabilities [4].
>
> **References:**
>
> [1] Kuhn, L., Gal, Y., & Farquhar, S. (2023). Semantic uncertainty: Linguistic invariances for uncertainty estimation in natural language generation. In Proceedings of the Eleventh International Conference on Learning Representations (ICLR).
>
> [2] Duan, J., Cheng, H., Wang, S., Zavalny, A., Wang, C., Xu, R., Kailkhura, B., & Xu, K. (2024). Shifting attention to relevance: Towards the predictive uncertainty quantification of free-form large language models. In Proceedings of the 62nd Annual Meeting of the Association for Computational Linguistics (Volume 1: Long Papers) (pp. 5050–5063). Association for Computational Linguistics.
>
> [3] Fadeeva, E., Rubashevskii, A., Shelmanov, A., Petrakov, S., Li, H., Mubarak, H., Tsymbalov, E., Kuzmin, G., Panchenko, A., Baldwin, T., Nakov, P., & Panov, M. (2024). Fact-checking the output of large language models via token-level uncertainty quantification. In Findings of the Association for Computational Linguistics: ACL 2024 (pp. 9367–9385). Association for Computational Linguistics.
>
> [4] ​​Vashurin, R., Fadeeva, E., Vazhentsev, A., Rvanova, L., Vasilev, D., Tsvigun, A., Petrakov, S., Xing, R., Sadallah, A., Grishchenkov, K., Panchenko, A., Baldwin, T., Nakov, P., Panov, M., & Shelmanov, A. (2025). Benchmarking uncertainty quantification methods for large language models with LM-Polygraph. Transactions of the Association for Computational Linguistics, 13, 220–248.

---

> ### Author Response · Authors · 2025-11-25
>
> Dear reviewer,
> Thank you again for your time and thoughtful feedback. With one week remaining in the discussion period, we kindly ask you to review our responses. Our rebuttal addresses the concerns you raised in your review. If there are no concerns remaining, we would greatly appreciate it if you could reconsider your scores accordingly. Should any questions remain, we would be happy to provide further clarification.

---

### Official Review · Reviewer_SuGS · 2025-11-01

**Soundness:** 3
**Presentation:** 4
**Contribution:** 3
**Rating:** 4
**Confidence:** 3

**Summary:**

They propose a new family of UQ methods based on beam search, using importance-weighted estimators to produce distinct candidate outputs. The authors provide a theoretical analysis alongwith an empirical analysis across multiple datasets and models.

**Strengths:**

- The paper is generally well written, well structured and easy to understand.

- It proposes a new way of leveraging beam search for UQ, going beyond routine sampling or decoding tweaks.

**Weaknesses:**

- The focus is primarily on short QA tasks. It’s not clear how well the approach would generalize to long-form or structured generation, or to tasks with less peaked probability distributions.

- An deeper study on beam width and similarity function for semantic entropy is missing.

- A more intuitive summary and visualization of the main theorem’s impact would increase accessibility.

**Questions:**

- How does the method scale to tasks involving longer generations, where the probability mass may be less concentrated on a few beams?

- How does it compare to multinomial sampling technique with averaging of confidence instead of just doing maximal voting?

- How to set beam width for different tasks or LLMs?

---

> ### Author Response · Authors · 2025-11-20
>
> We thank the reviewer for their constructive feedback. We address all concerns and suggestions in detail below.
>
> **Q1: The focus is primarily on short QA tasks. It’s not clear how well the approach would generalize to long-form or structured generation, or to tasks with less peaked probability distributions.**
>
>
> The proposed UQ method can in principle be used for both short- and long-form generations. However, our theoretical results and experimental findings indicate that it performs better on short answer QA. In the case of longer generations with autoregressive probabilistic models, less probability mass will be observed within a limited number of sequences generated by beam search. This in turn reduces the likelihood that total probability mass covered by beam search will exceed the threshold defined by Eq. 7. Empirical study of the prevalence of successful coverage is reported on Fig. 3, supporting this analysis. This explains why the performance gap between beam search and sampling approaches reduces as generations get longer, as indicated by Fig. 5.
>
> To further investigate performance of our approach on longer generations we also conduct experiments on the GSM8k dataset. We use reasoning elicitation with prompt, compute per-step uncertainty and aggregate it over steps via mean-pooling.
>
> | Method                                    | Value              |
> |-------------------------------------------|--------------------|
> | MSP                                       | .347               |
> | MTE                                       | .336               |
> | Perplexity                                | .332               |
> | CCP                                       | .303               |
> | P(True)                                   | .271               |
> | SemanticEntropy                           | .340               |
> | Lexical Similarity                        | .376               |
> | EigValLaplacian                           | .350               |
> | NumSemSets                                | .348               |
> | Dissimilarity                             | .354               |
> | Dissimilarity + beamsearch                | **.373**$\uparrow$ |
> | Eccentricity                              | .342               |
> | Eccentricity + beamsearch                 | .346$\uparrow$     |
> | EigVecDissimilarity                       | .337               |
> | EigVecDissimilarity + beamsearch          | .347$\uparrow$     |
> | CocoaMSP                                  | .360               |
> | CocoaMSP + beamsearch                     | .369$\uparrow$     |
> | CocoaPPL                                  | .337               |
> | CocoaPPL + beamsearch                     | .358$\uparrow$     |
>
>
> This supports our claim that beam search is generally preferable to multinomial sampling from the UQ standpoint, as it facilitates better UQ performance on short form tasks and works either better or identical to sampling on a long-form generation task.

---

> ### Author Response · Authors · 2025-11-20
>
> **Q2: An deeper study on beam width and similarity function for semantic entropy is missing.**
>
> To ensure we are aligned, we interpret your question as referring specifically to the Semantic Entropy baseline and its performance under different similarity functions ($s$) and beam widths ($M$).
>
> 1. **Beam Width**: We evaluated the performance of Semantic Entropy and Dissimilarity (both sampling and beam-search variants) across different beam widths $M \in$ {$1, …, 15$} on TriviaQA using Gemma3-4b-base (with the same NLI similarity function $s$) and included it in **Fig. 7 in appendix** of the updated paper PDF. The results show that for $M > 1$, Semantic Entropy underperforms compared to sampling-based Dissimilarity, while beam-search Dissimilarity yields an additional performance boost. As we discussed in the paper (Appendix A3), the underperformance of Semantic Entropy arises from a target mismatch: it measures uncertainty with respect to the input $x$ and is independent of the produced answer $y*$. In contrast, our primary methods, Dissimilarity, Eccentricity, and EigVecDissimilarity, directly assess the correctness of $y*$ itself.
>
> 2. **Similarity Function**: **Appendix A5** contains variation of our experimental results when NLI similarity is replaced with cross-encoder model trained on STS task. For convenience, we report a direct comparison of beamsearch-based methods with NLI vs STS similarity:
>
> | **UQ Method (beamsearch)** | TriviaQA | Web Questions | CoQA | HotpotQA | CommonsenseQA | ARC-Challenge |
> |---|---:|---:|---:|---:|---:|---:|
> | **Dissimilarity** |  |  |  |  |  |  |
> | NLI | **.766** | **.722** | **.600** | .611 | **.595** | **.604** |
> | Cross-encoder (STS) | .746 | .693 | .513 | **.654** | .505 | .479 |
> | **Eccentricity** |  |  |  |  |  |  |
> | NLI | **.739** | .633 | **.505** | .514 | **.590** | **.636** |
> | Cross-encoder (STS) | .734 | **.647** | .483 | **.604** | .362 | .421 |
> | **EigVecDissimilarity** |  |  |  |  |  |  |
> | NLI | **.753** | .668 | **.497** | .487 | **.562** | **.621** |
> | Cross-encoder (STS) | .744 | **.675** | .484 | **.582** | .439 | .496 |
> | **CocoaMSP** |  |  |  |  |  |  |
> | NLI | **.747** | **.679** | **.548** | **.523** | **.586** | **.606** |
> | Cross-encoder (STS) | .740 | .648 | .462 | .479 | .558 | .593 |
> | **CocoaPPL** |  |  |  |  |  |  |
> | NLI | **.748** | **.694** | **.577** | **.681** | **.582** | **.610** |
> | Cross-encoder (STS) | .737 | .658 | .498 | .650 | .548 | .586 |
>
> Generally, NLI similarity performs better (sometimes by huge margin). However on a few of the dataset-UQ method combinations the trend flips and STS shows better performance. These results suggest that NLI is a better default option for estimating semantic similarity of LLM outputs.
>
> **Semantic Entropy** was originally proposed with NLI similarity, and we followed this implementation, as well as used the same NLI similarity for other UQ methods in the main paper experiments. This ensures internal consistency of the main experimental results. While one could investigate how choice of similarity function affects performance of Semantic Entropy, we consider tweaking baseline methods to such a degree to be out of scope for this submission.
>
> **Q3: A more intuitive summary and visualization of the main theorem’s impact would increase accessibility.**
>
> Thank you for this helpful suggestion. We revised Section 3.2 (Theoretical Analysis) to better highlight and illustrate key ideas in the new paper PDF version.
>
> **Q4: How does it compare to multinomial sampling technique with averaging of confidence instead of just doing maximal voting?**
>
> To the best of our understanding, for both beamsearch and multinomial sampling, we do not do maximum voting or averaging of confidence for candidates. We have shown that using beam search candidates instead of multinomial sampling candidates improves performance and reduces variance for consistency-based UQ. Perhaps the reviewer could explain in more detail what exactly this comparison means in the given question?

---

> ### Author Response · Authors · 2025-11-20
>
> **Q5: How to set beam width for different tasks or LLMs?**
>
> While the main contribution of our work is the improved uncertainty estimation across any computational budget $M$, we agree that such ablations on different datasets and models would further strengthen the paper.
>
> We investigated the performance of the proposed estimators under varying beam widths $M$ for three LLMs (Gemma-4B, Llama-8B, Qwen-8B) and two datasets (TriviaQA and ARC-Challenge). We include the full results and discussion in the updated paper PDF, **Appendix A.6**.
>
> These experiments show that for open-ended generation tasks such as TriviaQA, the proposed beam-search variant of Dissimilarity consistently plateaus at a small budget of $M=5$ across all three LLMs. For multiple-choice tasks such as ARC-Challenge, the plateau occurs even earlier, around $M=2$, likely due to the small output space (i.e., a limited set of answer choices).
>
> Since these ablations require multiple additional runs per model-dataset pair, they are computationally expensive. For this reason, and given the consistent trends observed across the settings we evaluated, we defer producing the same plots for all remaining datasets and models to the camera-ready version.

---

> ### Author Response · Authors · 2025-11-25
>
> Dear reviewer,
> Thank you again for your time and thoughtful feedback. With one week remaining in the discussion period, we kindly ask you to review our responses. Our rebuttal addresses the concerns you raised in your review. If there are no concerns remaining, we would greatly appreciate it if you could reconsider your scores accordingly. Should any questions remain, we would be happy to provide further clarification.

---

### Official Review · Reviewer_mkAX · 2025-11-12

**Soundness:** 2
**Presentation:** 2
**Contribution:** 2
**Rating:** 0
**Confidence:** 4

**Summary:**

Uncertainty quantification is an important topic for LLMs. This paper focuses on the consistency-based methods that measure agreement among multiple generations. The authors argue that multinomial sampling often produces duplicate short answers and high run‑to‑run variance, especially in short‑form QA. They propose replacing samples with beam‑search candidates and computing importance‑weighted consistency scores over the beam set (top‑M). The goals were to reduce duplication, lower estimator variance and achieve efficiency and effectiveness with 'no extra cost', since beam search is already used for decoding.

**Strengths:**

**Interesting problem and good motivation**

> The paper pinpoints a concrete weakness of consistency‑based UQ in short QA: duplicates and variance from multinomial sampling, with evidence (e.g., duplicate rates 30–50% for 2–4 token outputs in TriviaQA with 10 samples) and intuitive illustrations (Fig. 1 and Fig. 2). The proposed beam‑weighted estimator is simple and broadly applicable across prior consistency‑based methods.

**Theoretical analysis**

> The paper provided comparison analysis, covering the MSE of the multinomial MC estimator (unbiased, variance) against the deterministic beam‑weighted estimator (bias from top‑M truncation, but no sampling variance). The resulting condition, for example beam mass $m_B$​ above a threshold (e.g., >0.842 for M=10), is interpretable and aligns with short‑form QA, where top few beams often capture most probability mass. This work shows this condition holds for a sizable subset and more often for short outputs.

**Empirical findings**

> Six datasets spanning closed‑book, open‑book, and multiple‑choice QA, six popular LLMs (base and instruct), and comparisons to a large set of information‑based and consistency‑based baselines implemented via LM‑Polygraph. The principal metric PRR (normalized AURC with AlignScore quality) follows recent UQ benchmarking recommendations.

**Weaknesses:**

**Limitation in theory**

> While the paper has provided comparison condition, it heavily rely on unknown “inside‑outside” gap and assumed similarity. The estimator is deterministic given a fixed beam set, but it still inherits the bias from top‑M truncation.

**Limitation in scope**

> Most gains are for short answers; the paper itself shows the advantage shrinks as outputs lengthen (Fig. 5). It is unclear whether the approach still helps long‑form generation (summarization, step‑by‑step reasoning), where beam search can become less diverse and costlier.

**Further report on cost claims**

> The paper states UQ is “essentially free” when beam search is already run; however, many UQ pipelines today do not decode with beam for generation (often nucleus/temperature sampling). The paper reports total GPU‑days, but not per‑query latency nor a direct throughput comparison between beam(M) and sampling(M) under matched compute.

**Sensitivity analysis**

> Although STS vs. NLI ablations are shown, results do shift on some datasets (Table 8). The paper also introduces a mass floor $\epsilon$ for stability, but the “best $\epsilon$” is case‑dependent (Table 5).

**Baselines selection**

> While diverse beam and temperature sampling are ablated, a natural question is how nucleus/temperature sampling with semantic deduplication (e.g., cluster‑then‑subsample) fares as a competing “low‑variance” sampler for short QA. The hybrid beam+sampling table suggests potential, but a semantic‑dedup sampling baseline would make the case stronger.

**Questions:**

Further on the questions raised in Weakness, please also answer my following questions.

1. Can you report per‑query latency/throughput comparisons for beam vs. multinomial sampling at the same MMM, both when (a) beam is not used for generation and when (b) it is used (to support the “free” claim)?

2. Beyond the sufficient bound, can you measure mBm_BmB​ and provide scatter plots of PRR gain vs.  $m_B$​? Any proxy to estimate the truncation bias on real data?

3. Do the gains persist for longer generations (e.g., multi‑sentence answers, summarization)? If not, why?

4. Can you include exact‑match (or normalized string match) for TriviaQA/WebQ and choice accuracy for MC to corroborate PRR conclusions that rely on AlignScore?

5. Given the sensitivity in Table 8, do you recommend NLI vs. STS depending on task type (factoid vs. conversational vs. MC)? How stable are results across different NLI models?

7. One claimed benefit is reduced run‑to‑run variance. Can you report std/CI of PRR over multiple runs for sampling‑ vs. beam‑based estimators at fixed M?

**Details Of Ethics Concerns:**

Public datasets

---

> ### Author Response · Authors · 2025-11-13
>
> We are grateful to the reviewer for the insightful comments provided in the review. However, we feel that there is a significant mismatch between provided critique and proposed rating (0), to the point where we feel the need to ask whether this was a misclick, or the rating was indeed intended. We respectfully ask reviewer to change the rating to a value more fitting to the comments, which seemingly do not invalidate our core contributions.

---

> > ### Comment · Reviewer_fdB8 · 2025-11-20
> >
> > I totally agree with the authors on this point.

---

> ### Author Response · Authors · 2025-11-20
>
> We thank the reviewer for their feedback. Our point-by-point responses are provided below.
>
> **Q1: While the paper has provided comparison condition, it heavily rely on unknown “inside‑outside” gap and assumed similarity. The estimator is deterministic given a fixed beam set, but it still inherits the bias from top‑M truncation.**
>
> We thank the reviewer for the comment and are happy to clarify these points.
>
> First, we respectfully disagree that our theoretical analysis relies heavily on the inside‑outside gap. As stated in Theorem 1, we also provide a **distribution-free sufficient condition that does not depend on the inside-outside gap** at all:
>
> $$ m_B > 1 - 1/2\sqrt{M} $$
>
> This condition depends only on the number of samples M and the beam mass $m_B$​. In the discussion following the theorem, we empirically show that this exact distribution-free condition is satisfied for a substantial fraction of examples (Fig. 3).
>
> Second, while the beam-weighted estimator does introduce bias, it simultaneously benefits from zero variance, unlike the Monte Carlo estimator. In other words, the estimator trades a small amount of bias for eliminating sampling variance. Our theoretical analysis quantifies this trade-off explicitly, and our empirical results (e.g., Table 3) consistently demonstrate that this bias-variance trade-off is beneficial: the beam-weighted estimator significantly outperforms MC sampling in practice.
>
> **Q2: Most gains are for short answers; the paper itself shows the advantage shrinks as outputs lengthen (Fig. 5). It is unclear whether the approach still helps long‑form generation (summarization, step‑by‑step reasoning), where beam search can become less diverse and costlier.**
>
> The proposed UQ method can in principle be used for both short- and long-form generations. However, our theoretical results and experimental findings indicate that it performs better on short answer QA. In the case of longer generations with autoregressive probabilistic models, less probability mass will be observed within a limited number of sequences generated by beam search. This in turn reduces the likelihood that total probability mass covered by beam search will exceed the threshold defined by Eq. 7. Empirical study of the prevalence of successful coverage is reported on Fig. 3, supporting this analysis. This explains why the performance gap between beam search and sampling approaches reduces as generations get longer, as indicated by Fig. 5.
>
> To further investigate performance of our approach on longer generations we also conduct experiments on the GSM8k dataset. We use reasoning elicitation with prompt, compute per-step uncertainty and aggregate it over steps via mean-pooling.
>
> | Method                                    | Value              |
> |-------------------------------------------|--------------------|
> | MSP                                       | .347               |
> | MTE                                       | .336               |
> | Perplexity                                | .332               |
> | CCP                                       | .303               |
> | P(True)                                   | .271               |
> | SemanticEntropy                           | .340               |
> | Lexical Similarity                        | .376               |
> | EigValLaplacian                           | .350               |
> | NumSemSets                                | .348               |
> | Dissimilarity                             | .354               |
> | Dissimilarity + beamsearch                | **.373**$\uparrow$ |
> | Eccentricity                              | .342               |
> | Eccentricity + beamsearch                 | .346$\uparrow$     |
> | EigVecDissimilarity                       | .337               |
> | EigVecDissimilarity + beamsearch          | .347$\uparrow$     |
> | CocoaMSP                                  | .360               |
> | CocoaMSP + beamsearch                     | .369$\uparrow$     |
> | CocoaPPL                                  | .337               |
> | CocoaPPL + beamsearch                     | .358$\uparrow$     |
>
>
> This supports our claim that beam search is generally preferable to multinomial sampling from the UQ standpoint, as it facilitates better UQ performance on short form tasks and works either better or identical to sampling on a long-form generation task.

---

> ### Author Response · Authors · 2025-11-20
>
> **Q3: The paper states UQ is “essentially free” when beam search is already run; however, many UQ pipelines today do not decode with beam for generation (often nucleus/temperature sampling). The paper reports total GPU‑days, but not per‑query latency nor a direct throughput comparison between beam(M) and sampling(M) under matched compute.**
>
> To compare beam search and multinomial sampling under matched compute, we measured per-query latency on 200 TriviaQA examples (10 samples vs 10 beams, batch size 10).
>
> Sampling required 2.144 s per query on average, while beam search required 2.179 s. Additionally, calculating Dissimilarity takes another 0.078 s for sampling, and 0.076 s for beamsearch on average.
>
> 1. We note that beam search produces stronger top-1 answers than sampling or greedy decoding, so in settings where beam decoding is already preferred for generation, our beamsearch-enhanced Dissimilarity is **~30 times faster** than original sampling-based Dissimilarity (2.144s + 0.078s for sampling vs 0.076s for beamsearch, as beams come “for free”).
>
> 2. Even if greedy was chosen as decoding strategy, and beamsearch is employed only for Uncertainty Quantification, our method introduces **only ~1.5% computational overhead** (2.179s + 0.076s for beamsearch vs 2.144s + 0.078s for sampling). Thus, the additional cost of using beam search in this case is minimal.
>
> **Q4: Although STS vs. NLI ablations are shown, results do shift on some datasets (Table 8).**
>
> The results in Table 8 still show that in the most cases (29 out of 35), the beam option outperforms its multinomial counterpart. For a more direct comparison we report the following results on STS vs NLI approach:
>
> | **UQ Method (beamsearch)** | TriviaQA | Web Questions | CoQA | HotpotQA | CommonsenseQA | ARC-Challenge |
> |---|---:|---:|---:|---:|---:|---:|
> | **Dissimilarity** |  |  |  |  |  |  |
> | NLI | **.766** | **.722** | **.600** | .611 | **.595** | **.604** |
> | Cross-encoder (STS) | .746 | .693 | .513 | **.654** | .505 | .479 |
> | **Eccentricity** |  |  |  |  |  |  |
> | NLI | **.739** | .633 | **.505** | .514 | **.590** | **.636** |
> | Cross-encoder (STS) | .734 | **.647** | .483 | **.604** | .362 | .421 |
> | **EigVecDissimilarity** |  |  |  |  |  |  |
> | NLI | **.753** | .668 | **.497** | .487 | **.562** | **.621** |
> | Cross-encoder (STS) | .744 | **.675** | .484 | **.582** | .439 | .496 |
> | **CocoaMSP** |  |  |  |  |  |  |
> | NLI | **.747** | **.679** | **.548** | **.523** | **.586** | **.606** |
> | Cross-encoder (STS) | .740 | .648 | .462 | .479 | .558 | .593 |
> | **CocoaPPL** |  |  |  |  |  |  |
> | NLI | **.748** | **.694** | **.577** | **.681** | **.582** | **.610** |
> | Cross-encoder (STS) | .737 | .658 | .498 | .650 | .548 | .586 |
>
> Generally, NLI similarity performs better (sometimes by huge margin). However on a few of the dataset-UQ method combinations the trend flips and STS shows better performance. These results suggest that NLI is a better default option for estimating semantic similarity of LLM outputs.
>
> **Q5: The paper also introduces a mass floor eps for stability, but the “best ” is case‑dependent (Table 5).**
>
> In Table 5 we stated that there is no single winner for the eps parameter for all datasets. For the main experiments we excluded the eps parameter (used eps=0), and all beam-based improvements are consistent across various datasets. For specific tasks, it may be useful to tune eps to get the best results.
>
> **Q6: While diverse beam and temperature sampling are ablated, a natural question is how nucleus/temperature sampling with semantic deduplication (e.g., cluster‑then‑subsample) fares as a competing “low‑variance” sampler for short QA. The hybrid beam+sampling table suggests potential, but a semantic‑dedup sampling baseline would make the case stronger.**
>
> The basic idea for our experiments was to perform comparison between beam search and sampling under compute parity and equal number of produced sequences conditions. While some sort of subsampling semantically diverse sequences from the larger set of sampled generations could show improvement over simple sampling, it would require violating one of the conditions. Either we must sample $N > M$ responses to get $M$ semantically diverse sequences, or compare simple sampling of M responses with a smaller set of diverse generations.
>
> Furthermore, semantic entropy can be viewed as a variant of such semantic subsampling, and we show (Table 6) that it also benefits from beam-based decoding.
>
> We  hope we understood correctly what you mean by semantic deduplication. If you had something else in mind could you please elaborate exactly what kind of procedure you refer to?

---

> ### Author Response · Authors · 2025-11-20
>
> **Q7: Do the gains persist for longer generations (e.g., multi‑sentence answers, summarization)? If not, why?**
>
> The proposed UQ method can in principle be used for both short- and long-form generations. However, our theoretical results and experimental findings indicate that it performs better on short answer QA. In the case of longer generations with autoregressive probabilistic models, less probability mass will be observed within a limited number of sequences generated by beam search. This in turn reduces the likelihood that total probability mass covered by beam search will exceed the threshold defined by Eq. 7. Empirical study of the prevalence of successful coverage is reported on Fig. 3, supporting this analysis. This explains why the performance gap between beam search and sampling approaches reduces as generations get longer, as indicated by Fig. 5.
>
> To further investigate performance of our approach on longer generations we also conduct experiments on the GSM8k dataset. We use reasoning elicitation with prompt, compute per-step uncertainty and aggregate it over steps via mean-pooling.
>
> | Method                                    | Value              |
> |-------------------------------------------|--------------------|
> | MSP                                       | .347               |
> | MTE                                       | .336               |
> | Perplexity                                | .332               |
> | CCP                                       | .303               |
> | P(True)                                   | .271               |
> | SemanticEntropy                           | .340               |
> | Lexical Similarity                        | .376               |
> | EigValLaplacian                           | .350               |
> | NumSemSets                                | .348               |
> | Dissimilarity                             | .354               |
> | Dissimilarity + beamsearch                | **.373**$\uparrow$ |
> | Eccentricity                              | .342               |
> | Eccentricity + beamsearch                 | .346$\uparrow$     |
> | EigVecDissimilarity                       | .337               |
> | EigVecDissimilarity + beamsearch          | .347$\uparrow$     |
> | CocoaMSP                                  | .360               |
> | CocoaMSP + beamsearch                     | .369$\uparrow$     |
> | CocoaPPL                                  | .337               |
> | CocoaPPL + beamsearch                     | .358$\uparrow$     |
>
>
> This supports our claim that beam search is generally preferable to multinomial sampling from the UQ standpoint, as it facilitates better UQ performance on short form tasks and works either better or identical to sampling on a long-form generation task.
>
> **Q8: Can you include exact‑match (or normalized string match) for TriviaQA/WebQ and choice accuracy for MC to corroborate PRR conclusions that rely on AlignScore?**
>
> Thank you for the suggestion. We use AlignScore as our primary metric because **prior work has shown it is better suited for UQ benchmarking** [1], as it captures semantic equivalence that exact match cannot (e.g., gold answer “6” vs. model output “six”).
>
> Still, to verify that our conclusions do not depend on AlignScore, we ran additional checks with Llama-8B-base, using normalized string match on TriviaQA and choice accuracy on ARC-Challenge (Multiple-Choice). The beam-search variants still outperform multinomial sampling, indicating that the improvements hold across quality metrics.
>
> | UQ Method                       | TriviaQA | ARC-Challenge |
> |---------------------------------|----------|----------------|
> | MSP                             | 0.635    | 0.520 |
> | MTE                             | 0.634    | 0.521 |
> | Perplexity                      | 0.616    | 0.585 |
> | CCP                             | 0.644    | 0.460 |
> | SAR                             | 0.608    | 0.340 |
> | P(True)                         | -0.032   | 0.295 |
> | SemanticEntropy                 | 0.605    | 0.521 |
> | Lexical Similarity              | 0.538    | 0.524 |
> | EigValLaplacian                 | 0.599    | 0.496 |
> | NumSemSets                      | 0.586    | 0.479 |
> | Dissimilarity                   | 0.620    | 0.648 |
> | Dissimilarity + beamsearch      | 0.690$\uparrow$ | **0.686**$\uparrow$ |
> | Eccentricity                    | 0.626    | 0.606 |
> | Eccentricity + beamsearch       | 0.691$\uparrow$ | 0.671$\uparrow$ |
> | EigVecDissimilarity             | 0.632    | 0.602 |
> | EigVecDissimilarity + beamsearch| 0.692$\uparrow$ | 0.636$\uparrow$ |
> | CocoaMSP                        | 0.670    | 0.646 |
> | CocoaMSP + beamsearch           | **0.703**$\uparrow$ | 0.656$\uparrow$ |
>
> [1] Vashurin et al. Benchmarking Uncertainty Quantification Methods for Large Language Models with LM-Polygraph. TACL

---

> ### Author Response · Authors · 2025-11-20
>
> **Q9: One claimed benefit is reduced run‑to‑run variance. Can you report std/CI of PRR over multiple runs for sampling‑ vs. beam‑based estimators at fixed M?**
>
> - We do claim “reduced variance” in both the abstract and conclusion, and we would like to clarify the distinction here. Our beam-based estimator is fully deterministic (beam search has no stochastic components), so it exhibits **zero run-to-run variance**. In contrast, the sampling-based Monte-Carlo Dissimilarity estimator is unbiased but has non-zero variance across runs.
>
> - As discussed in Section 3.2, beamsearch introduces bias, but the magnitude of this bias cannot be measured directly because of the exponential output space. Nonetheless, Theorem 1 characterizes conditions (commonly satisfied in real datasets, see Fig. 3), under which this bias is beneficial. Empirically, our results (e.g., Table 3) consistently show that beamsearch variations of all introduced methods improves accuracy compared to original sampling versions.
>
> - For reference, we computed variance over 10 seeds for Dissimilarity (M=10) on TriviaQA, Qwen3-8B-base. While the sampling estimator yields 0.668 ± 0.004 PRR across 10 seeds (probably because of the large dataset size: 2000 questions), the mean variance of underlying Dissimilarity scores across seeds is **0.183** (substantial for a [0,1] metric). In contrast, the beam-search variants are deterministic and exhibit **zero variance**, as expected.

---

> ### Author Response · Authors · 2025-11-25
>
> Dear reviewer,
> Thank you again for your time and thoughtful feedback. With one week remaining in the discussion period, we kindly ask you to review our responses. Our rebuttal addresses the concerns you raised in your review. If there are no concerns remaining, we would greatly appreciate it if you could reconsider your scores accordingly. Should any questions remain, we would be happy to provide further clarification.

---

### Meta-Review · Area_Chair_74Uy · 2025-12-11

**Summary:**

After skimming the paper myself and carefully familiarizing myself with the entirety of the Reviewer-Author discussion, I summarize the strengths and weaknesses identified by most reviewers as follows:

* Main strengths: A clean contribution, applicable to a wide range of multi-sample UQ methods. Applicability is clearly delineated with Theorem 1. Appears to improve the empirical performance of the estimates a little but consistently. Technically correct.
* Main weakness: The method is only beneficial for short-form NLG tasks (which is clearly acknowledged). Given most evaluation settings use few-shot examples (Table 1), the freeform responses generated are not even in the form of a sentence or of approximately sentence-length, they seem to be mostly 1-3 words long (judging by skimming the answers format for the first 3 datasets in Table 1 on https://huggingface.co/datasets). Hence, as per Theorem 1, I would expect that the already small improvement would decrease further if we were to evaluate this method on a sentence-length generations.

**Reviewer Concerns:**

Except for the main weakness above, all concerns were addressed.

**Reviewer Scores:**

Apart from Reviewer mkAX, whose score is an outlier, other reviewers seem to have identified similar strengths and weaknesses but weighted them differently in arriving at their final score (8, 8, 4, 4). fdB8 (8) and 1K6A (8) replied they would keep their positive score. jkE6 (4) and SuGS (4) would probably have increased their score after the rebuttal to 6.

This results in an overall score of 8, 8, 6, 6 which is well-above the acceptance bar and thus it is my pleasure to recommend acceptance for this submission.

---

### Decision · Program_Chairs · 2026-01-26

Accept (Poster)